# Heatmap Distribution Matching for Human Pose Estimation

**Haoxuan Qu**
SUTD
Singapore
haoxuan_qu@mymail.sutd.edu.sg

**Li Xu**
SUTD
Singapore
li_xu@mymail.sutd.edu.sg

**Yujun Cai**
NTU
Singapore
yujun001@e.ntu.edu.sg

**Lin Geng Foo**
SUTD
Singapore
lingeng_foo@mymail.sutd.edu.sg

**Jun Liu** *
SUTD
Singapore
jun_liu@sutd.edu.sg

## Abstract

For tackling the task of 2D human pose estimation, the great majority of the recent methods regard this task as a heatmap estimation problem, and optimize the heatmap prediction using the Gaussian-smoothed heatmap as the optimization objective and using the pixel-wise loss (e.g. MSE) as the loss function. In this paper, we show that optimizing the heatmap prediction in such a way, the model performance of body joint localization, which is the intrinsic objective of this task, may not be consistently improved during the optimization process of the heatmap prediction. To address this problem, from a novel perspective, we propose to formulate the optimization of the heatmap prediction as a distribution matching problem between the predicted heatmap and the dot annotation of the body joint directly. By doing so, our proposed method does not need to construct the Gaussian-smoothed heatmap and can achieve a more consistent model performance improvement during the optimization of the heatmap prediction. We show the effectiveness of our proposed method through extensive experiments on the COCO dataset and the MPII dataset.

## 1 Introduction

2D human pose estimation aims to locate body joints of a person in a given RGB image. It is relevant to a variety of applications, such as action recognition [34], human-machine interaction [40], and sign language understanding [19]. For tackling the task of 2D human pose estimation, most of the recent methods [29, 20, 33, 26, 14, 4, 39, 35, 15, 17, 37] are *heatmap-based*, i.e., they regard 2D human pose estimation as a heatmap estimation problem. Specifically, for each body joint, these methods generally estimate a grid-like heatmap, on which each pixel value represents the probability that this pixel contains the body joint. Compared to the methods [30, 2, 32, 13] that directly regress the coordinates of body joints (i.e. *coordinate regression-based methods*), the *heatmap-based methods*

---

*Corresponding Author

36th Conference on Neural Information Processing Systems (NeurIPS 2022).

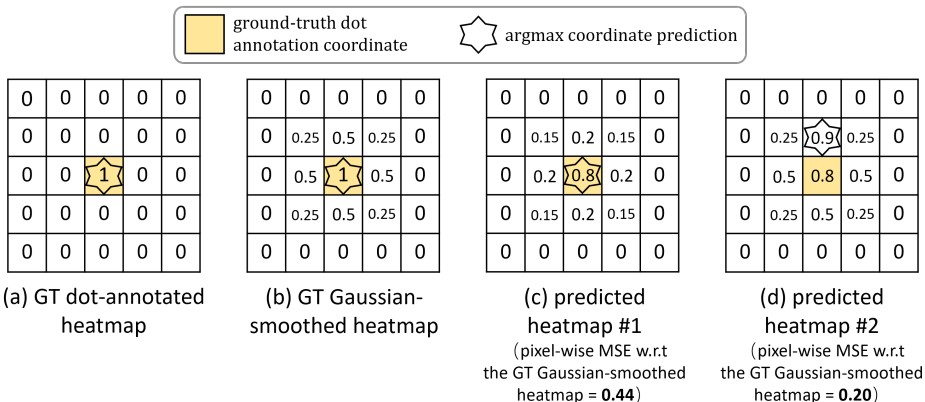

Figure 1: Illustration of heatmaps. Although the pixel-wise MSE loss calculated between the predicted heatmap #2 and the Gaussian-smoothed heatmap is smaller than the loss calculated between the predicted heatmap #1 and the Gaussian-smoothed heatmap, the predicted heatmap #2 localizes the body joint wrongly, whereas the predicted heatmap #1 localizes the body joint correctly.

demonstrate a more robust performance since they maintain the spatial structure of the input image throughout the encoding and decoding process [7].

During the training process of the *heatmap-based methods*, an important step is the optimization of the heatmap prediction. This optimization can be done naively via constructing a dot-annotated heatmap for each body joint as shown in Fig. 1(a), and then measuring the difference (i.e., conducting pixel-wise comparison) between the predicted heatmap and the constructed ground-truth (GT) dot-annotated heatmap. However, such a dot-annotated heatmap is sparse, as it has the same zero value for all pixels except the pixel representing the dot annotation of the body joint. Because of this, optimizing the heatmap prediction in such a naive way can lead to a hard training process and a suboptimal model performance [28]. To address this problem, most *heatmap-based methods* [29, 20, 33, 26, 14, 4, 39, 35, 15, 17, 37] adopt a strategy to construct Gaussian-smoothed heatmaps, where pixels near the dot annotation have larger pixel values than pixels far from the dot annotation. Specifically, they construct the Gaussian-smoothed heatmap via smoothing the dot annotation of the body joint through a Gaussian distribution as shown in Fig. 1(b), instead of only setting the pixel representing the dot annotation to be one.

While easing the training process, constructing the GT Gaussian-smoothed heatmap still brings problems into the model training process. Firstly, for constructing the Gaussian-smoothed heatmap, we need to choose a proper standard deviation of the Gaussian distributions. However, the proper standard deviations of the Gaussian distributions (i.e., the standard deviations that can lead to an optimal performance) often vary across different types of body joints, different body postures, and different body sizes [17]. Hence, the standard deviations of the Gaussian distributions often need to be carefully chosen, which is non-trivial. Secondly, during the process of optimizing the heatmap prediction by minimizing the pixel-wise loss (e.g. MSE) between the predicted heatmap and the Gaussian-smoothed heatmap, the model performance of body joint localization may not be consistently improved. As shown in Fig. 1, although compared to the loss calculated between the predicted heatmap #1 and the Gaussian-smoothed heatmap, the pixel-wise MSE loss calculated between the predicted heatmap #2 and the Gaussian-smoothed heatmap is smaller, the predicted heatmap #2 localizes the body joint wrongly, whereas the predicted heatmap #1 localizes the body joint correctly.

As a result, optimizing the heatmap prediction using the dot-annotated heatmap and the Gaussian-smoothed heatmap as the ground-truth both have their respective problems. Hence, in this work, we aim to tackle their respective problems, and propose to optimize the heatmap prediction directly via minimizing the difference between the predicted heatmap and the dot annotation. By doing so, we can optimize the model directly towards accurately localizing the dot annotation of the body joint, which is the intrinsic objective of 2D human pose estimation, instead of optimizing the model indirectly towards either the dot-annotated heatmap or the Gaussian-smoothed heatmap. However, as the number of pixels in the predicted heatmap and the number of entries representing the dot

annotation are different, we cannot measure the difference between the predicted heatmap and the dot annotation trivially by measuring their entry-wise difference. To handle this problem, inspired by the fact that we can measure the difference between two distributions via measuring their Earth Mover's Distance even if they have different numbers of entries, in this paper, we propose to first formulate the optimization of the heatmap prediction as a distribution matching problem. Specifically, we construct two distributions respectively from the predicted heatmap and the dot annotation. After that, we optimize the heatmap prediction via minimizing the distribution difference based on the Earth Mover's Distance. Using such a novel method to optimize the heatmap prediction directly from the dot annotation, we do not need to construct the Gaussian-smoothed heatmap, as well as avoiding the issues of the binary dot-annotated heatmap. Thus, our method achieves superior performance.

Our proposed method is simple yet effective, which can be easily applied to various off-the-shelf 2D human pose estimation models by replacing their original loss function with our proposed loss function measuring the distribution difference between the predicted heatmap and the dot annotation. We experiment our proposed method on multiple models and our method achieves a consistent model performance improvement.

The contributions of our work are summarized as follows. 1) We analyze (in Sec. 4) that the performance of the human pose estimation model may not be consistently improved during the process of minimizing the pixel-wise loss between the predicted heatmap and the GT Gaussian-smoothed heatmap. 2) From a novel perspective, we formulate the optimization of the heatmap prediction as a distribution matching problem between the predicted heatmap and the GT dot annotation directly, which bypasses the step of constructing the Gaussian-smoothed heatmap and achieves consistent model performance improvement. 3) Our proposed method achieves state-of-the-art performance on the evaluated benchmarks.

## 2 Related Work

**2D Human Pose Estimation.** Due to the wide range of applications, the task of 2D human pose estimation has received lots of attention [30, 2, 32, 13, 21, 29, 20, 33, 26, 14, 4, 39, 35, 15, 17, 37, 27, 7, 8]. DeepPose [30] made the first attempt of applying deep neural networks into the task of 2D human pose estimation via directly regressing the coordinates of body joints. This type of *coordinate regression-based methods* [30, 2, 32, 13] often show inferior performances compared to the *heatmap-based methods* [29, 20, 33, 26, 14, 4, 39, 35, 15, 17, 37], as the *heatmap-based methods* can preserve the spatial structure of the input image throughout the encoding and decoding process [7]. Hence, recently, the great majority of the state-of-the-art methods [29, 20, 33, 26, 14, 4, 39, 35, 15, 17, 37] regard 2D human pose estimation as a heatmap estimation problem instead of the coordinate regression problem. Among the *heatmap-based methods*, Tompson et al. [29] proposed to apply Markov Random Field (MRF) into the task of 2D human pose estimation. After that, an "hourglass" network, with a conv-deconv architecture, was proposed by Newell et al. [20]. Xiao et al. [33] proposed a baseline method to predict the heatmap via adding several deconvolutional layers to a backbone network. Later on, to maintain high-resolution representations throughout the heatmap estimation process, HRNet was proposed by Sun et al. [26]. Yuan et al. [37] further proposed HRFormer to learn the high-resolution representations utilizing a transformer-based architecture. Besides the above *heatmap-based methods* that use the Gaussian-smoothed heatmap as the optimization objective, there are also some methods [27, 7, 8] that combine the idea of *heatmap* and *coordinate regression* by taking the expectation of the predicted heatmap as the predicted coordinates.

Here in this work, different from previous works, our method bypasses both the step of regression and the step of constructing the Gaussian-smoothed heatmap as the optimization objective. Instead, from a novel perspective, we propose to formulate the optimization of the heatmap prediction as a distribution matching problem by minimizing the distribution difference between the predicted heatmap and the dot annotation.

**Distribution Matching.** The idea of distribution matching has been studied in various tasks [24, 25, 31, 38, 41, 23], such as image retrieval [24], tracking [25], few-shot learning [38], and long-tail recognition [23]. In this work, from a novel perspective, we design a new distribution matching scheme to optimize the heatmap prediction with the help of sub-pixel resolutions for 2D human pose estimation.

# 3 Method

In 2D human pose estimation, optimizing the heatmap prediction using the dot-annotated heatmap and the Gaussian-smoothed heatmap as the ground-truth both have their respective problems. Hence, in this work, we aim to handle their respective problems, and optimize the heatmap prediction directly with the dot annotation of the body joint. To achieve this goal, we propose to formulate the optimization of the heatmap prediction as a distribution matching problem, and minimize the difference between the distribution constructed from the predicted heatmap and that constructed from the dot annotation based on the Earth Mover's Distance.

Below, we first give a brief review of the Earth Mover's Distance, and then discuss how we formulate the heatmap optimization process as a distribution matching problem. After that, we introduce how we construct the loss function measuring the distribution difference.

## 3.1 Revisiting Earth Mover's Distance

The Earth Mover's Distance is a a technique used for measuring the difference between two probability distributions, which can be understood as the optimal cost needed to transport the mass from one distribution to another. Specifically, for calculating the Earth Mover's Distance, we regard the source distribution as a set of ($N$) suppliers $S = (s_1, ..., s_N)^\top$, where $s_n$ represents the total units of mass that the $n$-th supplier has, and we regard the target distribution as a set of ($M$) demanders $D = (d_1, ..., d_M)^\top$, where $d_m$ represents the total units of mass that the $m$-th demander requires. Besides, we also denote $C \in R_{\geq 0}^{N \times M}$ as the cost function between the source and target distributions, where $C_{n,m}$ represents the cost for transporting a unit of mass from the $n$-th supplier to the $m$-th demander. Then we aim to find a least-cost transportation plan from the set of possible plans $P = \{p \in R_{\geq 0}^{N \times M} : p\mathbf{1} = S \ \& \ p^\top \mathbf{1} = D\}$ to transport all mass from the $N$ suppliers to the $M$ demanders, where $\mathbf{1}$ represents a vector of ones. The Earth Mover's Distance $E_C(S, D)$ denotes the cost of the least-cost transportation plan, which can be formulated as:

$$E_C(S, D) = \min_{p \in P} \ \langle C, p \rangle \tag{1}$$

where $\langle \cdot, \cdot \rangle$ represents the Frobenius dot product.

However, optimizing Eq. 1 directly is computationally expensive. Hence, to reduce the computational cost especially when handling large-scale problems, Cuturi [5] proposed a regularized formulation of the Earth Mover's Distance $E_C^{reg}(S, D)$ as:

$$E_C^{reg}(S, D) = \langle C, p^{reg} \rangle \ \textbf{where} \ p^{reg} = \underset{p \in P}{\arg\min} \ \left[ \langle C, p \rangle - \frac{1}{\lambda} h(p) \right] \tag{2}$$

where $\lambda > 0$, and $h(p) = -\sum_{n=1}^{N} \sum_{m=1}^{M} C_{n,m} \log C_{n,m}$. Optimizing Eq. 2 is computationally cheaper than optimizing Eq. 1, since Eq. 2 can be optimized with matrix scaling through the Sinkhorn algorithm.

## 3.2 Problem Formulation

Below, we formulate the optimization of the heatmap prediction as a distribution matching problem by constructing the suppliers $S$, the demanders $D$, and the cost function $C$.

**The suppliers $S$.** For a body joint, let $H_{pred} \in R^{H \times W}$ denote its corresponding predicted heatmap with height $H$ and width $W$. To formulate the suppliers $S$ to represent $H_{pred}$, we first localize a set of $H \times W$ suppliers corresponding to the $H \times W$ pixels of $H_{pred}$. Then, for determining the units of mass each supplier stores, as the suppliers cannot hold negative units of mass, we construct $S$ based on a non-nagetive formulation of $H_{pred}$. Besides, we also constrain the total units of mass stored by the suppliers $S$ to be the same as the total units of mass required by the demanders $D$. To meet these requirements, we derive $S$ by first passing $H_{pred}$ through a $relu$ activation function and then normalizing it as:

$$S = \frac{relu(H_{pred})}{\|relu(H_{pred})\|_1} \tag{3}$$

**The demanders $D$.** As for the demanders, we aim to construct $D$ w.r.t. each body joint to represent its corresponding GT dot annotation. To achieve this goal, for each body joint, a naive formulation of

$D$ is to identify the pixel containing the GT dot annotation (i.e., the upper left pixel in Fig. 2(a)) and construct a single demander (i.e., the yellow dot in Fig. 2(a)) at the center of this pixel. However, this naive formulation can result in a suboptimal model performance, as the demander formulated in this way can be noticeably different from what the GT dot annotation might suggest. Generally, the predicted heatmaps outputted by most of the existing *heatmap-based methods* [29, 20, 33, 26, 14, 4, 39, 35, 15, 17, 37] have a lower resolution compared to the input image. For example, for the method HRNet [26], when the size of the input image is $384 \times 288$, the size of the predicted heatmap is $96 \times 72$ only. Due to such a resolution gap, as shown in Fig. 2(a), there can exist a non-negligible distance between the location of the demander and the location of the dot annotation, which can affect the performance of the pose estimation model.

To address this problem, we aim to make the formulated demanders $D$ a more accurate representation of the dot annotation by overcoming the resolution gap. To achieve this, inspired by the fact that the more accurate location information of an object can be deduced using sub-pixel resolutions [12], we propose to formulate the demanders $D$ with the following four steps. (1) As illustrated in the upper left of Fig. 2(b), we first split the pixel containing the dot annotation into four "sub-pixels" (four squares separated from each other by dashed lines). (2) After that, we identify the "sub-pixel" that the dot annotation lies in (i.e., the green "sub-pixel" shown in Fig. 2(b)). (3) Then, as shown by the four yellow dots in Fig. 2(b), we localize a set of four demanders at the centers of both the pixel containing the identified "sub-pixel" (i.e., the upper left pixel in Fig. 2(b)), and the three pixels adjacent to the identified "sub-pixel" (i.e., the three blue pixels in Fig. 2(b)). (4) Finally, we determine the units of mass each demander requires in the following way so that the dot annotation can be precisely represented using the set of the four demanders. Specifically, denote the 2D Euclidean distance between the centers of two adjacent pixels as $g$ (which is also the pixel size), the coordinates of the GT dot annotation as $(x_{dot}, y_{dot})$, and the coordinates of the $i$-th demander as $(x_i, y_i)$, where $i \in \{1, 2, 3, 4\}$. Then we construct the demanders $D$ as:

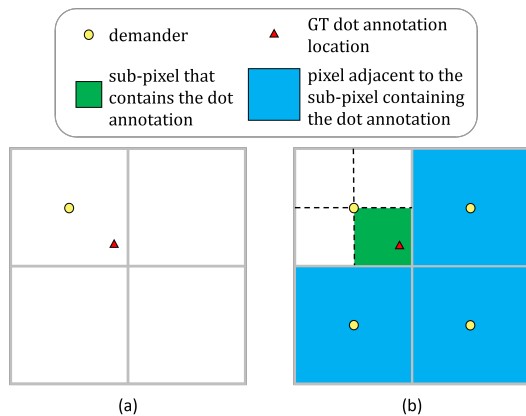
Figure 2: Illustration of (a) the naive formulation of the demanders $D$, and (b) formulation of the demanders $D$ that involves the idea of sub-pixel resolution.

$$D = (d_1, d_2, d_3, d_4), \textbf{ where } d_i = \frac{(g - |x_{dot} - x_i|) \times (g - |y_{dot} - y_i|)}{g^2} \qquad (4)$$

where $d_i$ denotes the units of mass the $i$-th demander requires. By designing $d_i$ in this way, we can achieve the following two properties. (1) Among the four demanders, via assigning more mass to the demanders nearer the dot annotation and less mass to the demanders farther from the dot annotation, we can accurately derive the coordinates of the dot annotation as $(x_{dot}, y_{dot}) = (d_1 \times x_1 + d_2 \times x_2 + d_3 \times x_3 + d_4 \times x_4, d_1 \times y_1 + d_2 \times y_2 + d_3 \times y_3 + d_4 \times y_4)$. Hence, the set of demanders $D$ constructed in this way can represent the GT dot annotation accurately, regardless of how many times the resolution of the heatmap is lower than the resolution of the input image. (2) Besides, as $d_1 + d_2 + d_3 + d_4 = 1$, the total units of mass required by the demanders $D$ are also constrained to be the same as the total units of mass stored by the suppliers $S$.

**The cost function $C$.** To measure the distribution difference between the $H \times W$ suppliers ($S$) constructed from the predicted heatmap and the 4 demanders ($D$) constructed from the GT dot annotation via calculating their Earth Mover's Distance, we also need to formulate a cost function $C \in R_{\geq 0}^{HW \times 4}$. Denote the coordinates of the $n$-th supplier as $(x_{s_n}, y_{s_n})$, where $n \in \{1, ..., H \times W\}$, and the coordinates of the $m$-th demander as $(x_{d_m}, y_{d_m})$, where $m \in \{1, 2, 3, 4\}$. We here simply formulate $C_{n,m}$, the cost per unit transported from the $n$-th supplier to the $m$-th demander, as the L2 distance between the $n$-th supplier and the $m$-th demander, i.e., $C_{n,m} = \sqrt{(x_{d_m} - x_{s_n})^2 + (y_{d_m} - y_{s_n})^2}$. Using such a cost function, our method can optimize the model directly towards accurately localizing the dot annotation of the body joint via minimizing the distribution difference between the suppliers $S$ and the demanders $D$.

### 3.3 Loss Function

Above we formulate the optimization of the heatmap prediction as a distribution matching problem for a single body joint. In this section, we introduce how we construct the loss function following such a formulation. Note that in 2D human pose estimation, we need to locate multiple body joints. Therefore, to construct the loss function, we first construct the suppliers, the demanders, and the cost function for each body joint respectively. After that, we calculate the loss value corresponding to each body joint via measuring the distribution difference (calculating the Earth Mover's Distance) between its corresponding demanders and suppliers. Specifically, denote $K$ the total number of body joints and $L_k$ the loss term calculated w.r.t. the $k$-th body joint. We then formulate the loss function as:

$$L_{Matching} = \sum_{k=1}^{K} L_k, \textbf{ where } L_k = E_{C^k}^{reg}(S^k, D^k) \tag{5}$$

where $S^k$, $D^k$, and $C^k$ respectively denote the constructed suppliers, demanders, and cost function for the $k$-th body joint. The corresponding Earth Mover's Distance $E_{C^k}^{reg}(S^k, D^k)$ is calculated using Eq. 2.

### 3.4 Training and Testing

Our proposed method can be flexibly applied on various off-the-shelf 2D human pose estimation models. During training, we optimize the heatmap prediction via minimizing the loss function in Eq. 5. During testing, for each body joint, we first select a square of four adjacent pixels with the largest sum of pixel values from the predicted heatmap, and then normalize the sum of these four pixel values to 1. The normalized pixel values of these four pixels can be seen as the units of mass each of them requires. Hence, in the same way as how we get the coordinates of the GT dot annotation from the demanders in Sec. 3.2, we can get the predicted coordinates of this joint. Specifically, denote the coordinates of the $i$-th selected pixel as $(x_i, y_i)$, and the normalized pixel value of this pixel as $d_i$, where $i \in \{1, 2, 3, 4\}$. We can derive the predicted coordinates of this joint as $(x_{pred}, y_{pred}) = (d_1 \times x_1 + d_2 \times x_2 + d_3 \times x_3 + d_4 \times x_4, d_1 \times y_1 + d_2 \times y_2 + d_3 \times y_3 + d_4 \times y_4)$.

## 4 Analysis

Most of the *heatmap-based methods* [29, 20, 33, 26, 14, 4, 39, 35, 15, 17, 37] optimize the heatmap prediction via minimizing the pixel-wise MSE loss between the predicted heatmap and the Gaussian-smoothed heatmap. Below we do some analysis about this type of methods.

We denote $K$ the number of joints per input image $I$. Then we denote $\mathbf{H_{dot}} = \{H_{dot}^1, ..., H_{dot}^K\}$, $\mathbf{H_{Gau}} = \{H_{Gau}^1, ..., H_{Gau}^K\}$, and $\mathbf{H_{pred}} = \{H_{pred}^1, ..., H_{pred}^K\}$ respectively the corresponding $K$ GT dot-annotated heatmaps, GT Gaussian-smoothed heatmaps, and predicted heatmaps of the input image $I$. We denote $\mathcal{D}_{dot} = \{(I, \mathbf{H_{dot}})\}$ the joint distribution of the input image and the corresponding $K$ dot-annotated heatmaps, and $\mathcal{D}_{Gau} = \{(I, \mathbf{H_{Gau}})\}$ the joint distribution of the input image and the corresponding $K$ Gaussian-smoothed heatmaps. Besides, we denote $l_{MSE}(a, b) = \|a - b\|_2^2$ the pixel-wise MSE loss, and $\phi$ the model parameters where $\phi(I) = \mathbf{H_{pred}}$. After that, we denote $R(\mathcal{D}_{dot}, \phi, l_{MSE}) = \mathbb{E}_{(I, \mathbf{H_{dot}}) \sim \mathcal{D}_{dot}}[l_{MSE}(\phi(I), \mathbf{H_{dot}})]$ as the expected risk calculated between the predicted heatmaps and the GT dot-annotated heatmaps, and $R(\mathcal{D}_{Gau}, \phi, l_{MSE}) = \mathbb{E}_{(I, \mathbf{H_{Gau}}) \sim \mathcal{D}_{Gau}}[l_{MSE}(\phi(I), \mathbf{H_{Gau}})]$ as the expected risk calculated between the predicted heatmaps and the Gaussian-smoothed heatmaps.

**Theorem 1.** *The relationship between $R(\mathcal{D}_{dot}, \phi, l_{MSE})$ and $R(\mathcal{D}_{Gau}, \phi, l_{MSE})$ can be written as*

$$\begin{aligned} R(\mathcal{D}_{dot}, \phi, l_{MSE}) = & R(\mathcal{D}_{Gau}, \phi, l_{MSE}) + 2 \times \mathbb{E}_{(I, \mathbf{H_{Gau}}) \sim \mathcal{D}_{Gau}}[\langle \mathbf{H_{pred}}, \mathbf{H_{Gau}} \rangle] \\ & - 2 \times \mathbb{E}_{(I, \mathbf{H_{dot}}) \sim \mathcal{D}_{dot}}[\langle \mathbf{H_{pred}}, \mathbf{H_{dot}} \rangle] - C \end{aligned} \tag{6}$$

*where $C = \mathbb{E}_{(I, \mathbf{H_{dot}}) \sim \mathcal{D}_{dot}}[\|\mathbf{H_{Gau}} - \mathbf{H_{dot}}\|_2^2]$ is a constant.*

The proof of Theorem 1 is provided in the supplementary. As shown in Theorem 1, when we minimize the pixel-wise loss between $\mathbf{H_{pred}}$ and $\mathbf{H_{Gau}}$, while $R(\mathcal{D}_{Gau}, \phi, l_{MSE})$ decreases, the second term $2 \times \mathbb{E}_{(I, \mathbf{H_{Gau}}) \sim \mathcal{D}_{Gau}}[\langle \mathbf{H_{pred}}, \mathbf{H_{Gau}} \rangle]$ can increase and the third term $2 \times \mathbb{E}_{(I, \mathbf{H_{dot}}) \sim \mathcal{D}_{dot}}[\langle \mathbf{H_{pred}}, \mathbf{H_{dot}} \rangle]$ cannot be guaranteed to increase or decrease. Because of this,

Table 1: The improvement of AP on COCO validation set when our proposed method is applied to various baselines.

| Method | Venue | Backbone | Input size | AP | $AP^{50}$ | $AP^{75}$ | $AP^M$ | $AP^L$ | AR |
|---|---|---|---|---|---|---|---|---|---|
| Hourglass[20] | ECCV 2016 | 8-Stage Hourglass | $256 \times 192$ | 66.9 | - | - | - | - | - |
| CPN[3] | CVPR 2018 | ResNet-50 | $256 \times 192$ | 69.4 | - | - | - | - | - |
| CPN[3] | CVPR 2018 | ResNet-50 | $384 \times 288$ | 71.6 | - | - | - | - | - |
| DarkPose[39] | CVPR 2020 | HRNet-W32 | $384 \times 288$ | 76.6 | 90.7 | 82.8 | 72.7 | 83.9 | 81.5 |
| UDP[10] | CVPR 2020 | HRNet-W32 | $384 \times 288$ | 77.8 | 91.7 | 84.5 | 74.2 | 84.3 | 82.4 |
| UDP[10] | CVPR 2020 | HRNet-W48 | $384 \times 288$ | 77.8 | 92.0 | 84.3 | 74.2 | 84.5 | 82.5 |
| TokenPose[15] | ICCV 2021 | TokenPose-L/D24 | $256 \times 192$ | 75.8 | 90.3 | 82.5 | 72.3 | 82.7 | 80.9 |
| Removing Bias[7] | ICCV 2021 | ResNet-152 | $384 \times 288$ | 74.4 | - | - | - | - | - |
| Removing Bias[7] | ICCV 2021 | HRNet-W32 | $256 \times 192$ | 75.8 | - | - | - | - | - |
| Simple Baseline[33] | ECCV 2018 | ResNet-152 | $384 \times 288$ | 75.0 | 90.8 | 82.1 | 67.8 | 78.3 | 80.0 |
| **+ Ours** | | ResNet-152 | $384 \times 288$ | **76.7(↑1.7)** | **92.1** | **83.6** | **69.7** | **80.0** | **81.3** |
| HRNet[26] | CVPR 2019 | HRNet-W32 | $384 \times 288$ | 76.7 | 91.9 | 83.6 | 73.2 | 83.2 | 81.6 |
| **+ Ours** | | HRNet-W32 | $384 \times 288$ | **78.2(↑1.5)** | **92.2** | **84.5** | **74.3** | **84.7** | **82.5** |
| HRNet[26] | CVPR 2019 | HRNet-W48 | $384 \times 288$ | 77.1 | 91.8 | 83.8 | 73.5 | 83.5 | 81.8 |
| **+ Ours** | | HRNet-W48 | $384 \times 288$ | **78.8(↑1.7)** | **92.5** | **85.1** | **75.0** | **85.3** | **83.1** |
| HRFormer[37] | NIPS 2021 | HRFormer-Base | $384 \times 288$ | 78.0 | 92.2 | 84.8 | 74.3 | 84.6 | 82.6 |
| **+ Ours** | | HRFormer-Base | $384 \times 288$ | **78.9(↑0.9)** | **92.6** | **85.4** | **75.3** | **85.3** | **83.3** |

$R(\mathcal{D}_{dot}, \phi, l_{MSE})$ cannot be guaranteed to decrease, and the model performance of body joint localization may not be consistently improved during such optimization of heatmap prediction. A more intuitive analysis is as follows. In the optimization process of most of the *heatmap-based methods*, since all the $H \times W$ pixels from the predicted heatmap contribute to the overall pixel-wise loss, learning to fit the other pixels better instead of the pixel representing the dot annotation can also lead to a smaller overall loss, as shown in Fig. 1. Hence, when minimizing the overall pixel-wise loss, the model performance of body joint localization cannot be guaranteed to improve consistently. Besides, during training, the GT Gaussian-smoothed heatmap is constructed via using the Gaussian blob. Therefore, during testing, the predicted heatmap often has a relatively large area of pixels with large values around the dot annotation, as shown in Fig. 3, which can make it difficult to accurately locate the body joint.

Differently, our method optimizes the heatmap prediction from a novel perspective via minimizing the difference between the distribution constructed from the predicted heatmap (i.e., the suppliers $S$), and the distribution constructed from the dot annotation (i.e., the demanders $D$). During training, since we formulate the cost function between each pair of supplier and demander as their L2 distance, minimizing the distribution difference based on such a cost function can aggregate the pixel values in the predicted heatmap towards the dot annotation. Hence, our method can help to achieve a more consistent model performance improvement when minimizing the distribution difference. Besides, during testing, equipped with such a mechanism of aggregating the predicted pixel values towards the dot annotation, our method can achieve a more compact body joint localization as shown in Fig. 3.

## 5 Experiments

To evaluate the effectiveness of our proposed method, we conduct experiments on the COCO dataset [16] and the MPII Human Pose dataset [1]. Besides, to test the generality of our method, we apply it to various backbones, e.g., ResNet [33], HRNet [26], and HRFormer [37].

### 5.1 COCO Keypoint Detection

**Dataset & evaluation metric.** The COCO dataset [16] contains more than 200k images and 250k person instances, which are annotated with 17 body joints. This dataset has three subsets including COCO training set, COCO validation set, and COCO test-dev set, which have 57k, 5k and 20k images, respectively. We conduct experiments on this dataset via first training the model on the train2017 set, and then evaluating the model on the val2017 set and test-dev2017 set. Following [33, 26, 37], we use standard average precision (AP) calculated based on Object Keypoint Similarity (OKS) to evaluate model performance.

**Implementation details.** We apply our method to various baselines including Simple Baseline [33], HRNet [26], and HRFormer [37], with their respective backbones including ResNet-152, HRNet-W32, HRNet-W48, and HRFormer-Base. For these baselines, we follow their original learning and

Table 2: The improvement of AP on COCO test-dev set when our proposed method is applied to various baselines.

| Method | Venue | Backbone | Input size | AP | AP$^{50}$ | AP$^{75}$ | AP$^M$ | AP$^L$ | AR |
|---|---|---|---|---|---|---|---|---|---|
| G-RMI[22] | CVPR 2017 | ResNet-101 | $353 \times 257$ | 64.9 | 85.5 | 71.3 | 62.3 | 70.0 | 69.7 |
| Mask-RCNN[9] | ICCV 2017 | ResNet-50-FPN | - | 63.1 | 87.3 | 68.7 | 57.8 | 71.4 | - |
| RMPE[6] | ICCV 2017 | PyraNet[36] | $320 \times 256$ | 72.3 | 89.2 | 79.1 | 68.0 | 78.6 | - |
| CFN[11] | ICCV 2017 | - | - | 72.6 | 86.1 | 69.7 | 78.3 | 64.1 | - |
| CPN[3] | CVPR 2018 | ResNet-Inception | $384 \times 288$ | 72.1 | 91.4 | 80.0 | 68.7 | 77.2 | 78.5 |
| CPN(ensemble)[3] | CVPR 2018 | ResNet-Inception | $384 \times 288$ | 73.0 | 91.7 | 80.9 | 69.5 | 78.1 | 79.0 |
| Integral Pose Regression[27] | ECCV 2018 | ResNet-101 | $256 \times 256$ | 67.8 | 88.2 | 74.8 | 63.9 | 74.0 | - |
| Posefix[18] | CVPR 2019 | ResNet-152 | $384 \times 288$ | 73.6 | 90.8 | 81.0 | 70.3 | 79.8 | 79.0 |
| DarkPose[39] | CVPR 2020 | HRNet-W48 | $384 \times 288$ | 76.2 | 92.5 | 83.6 | 72.5 | 82.4 | 81.1 |
| UDP[10] | CVPR 2020 | HRNet-W48 | $384 \times 288$ | 76.5 | 92.7 | 84.0 | 73.0 | 82.4 | 81.6 |
| TokenPose[15] | ICCV 2021 | TokenPose-L/D24 | $384 \times 288$ | 75.9 | 92.3 | 83.4 | 72.2 | 82.1 | 80.8 |
| Removing Bias[7] | ICCV 2021 | HRNet-W48 | $384 \times 288$ | 76.1 | - | - | - | - | 81.0 |
| Simple Baseline[33] | ECCV 2018 | ResNet-152 | $384 \times 288$ | 73.8 | 91.7 | 81.2 | 70.3 | 80.0 | 79.1 |
| **+ Ours** | | ResNet-152 | $384 \times 288$ | **75.3**(↑**1.5**) | **92.6** | **83.1** | **71.7** | **81.1** | **80.3** |
| HRNet[26] | CVPR 2019 | HRNet-W32 | $384 \times 288$ | 74.9 | 92.5 | 82.8 | 71.3 | 80.9 | 80.1 |
| **+ Ours** | | HRNet-W32 | $384 \times 288$ | **76.7**(↑**1.8**) | **92.6** | **84.0** | **73.0** | **82.8** | **81.5** |
| HRNet[26] | CVPR 2019 | HRNet-W48 | $384 \times 288$ | 75.5 | 92.5 | 83.3 | 71.9 | 81.5 | 80.5 |
| **+ Ours** | | HRNet-W48 | $384 \times 288$ | **77.2**(↑**1.7**) | **93.0** | **84.4** | **73.4** | **83.3** | **82.0** |
| HRFormer[37] | NIPS 2021 | HRFormer-Base | $384 \times 288$ | 76.2 | 92.7 | 83.8 | 72.5 | 82.3 | 81.2 |
| **+ Ours** | | HRFormer-Base | $384 \times 288$ | **77.2**(↑**1.0**) | **93.1** | **84.7** | **73.8** | **83.0** | **82.1** |

optimization configurations for model training. To calculate the Earth Mover's Distance using the Sinkhorn algorithm, we set the Sinkhorn entropic regularization parameter to 1 and the number of Sinkhorn iterations to 1000 in our experiments.

**Results.** In Tab. 1 and Tab. 2, we report results on the COCO validation and test-dev sets. We observe that after applying our method on various baselines, a significant performance enhancement is achieved, which shows effectiveness of our proposed method. Moreover, we compare our method with other state-of-the-art 2D human pose estimation methods. Our method achieves superior performance compared to these methods, further demonstrating the effectiveness of our method.

## 5.2 MPII Human Pose Estimation

**Dataset & evaluation metric.** The MPII dataset [1] contains around 25K images and more than 40k person instances, which are annotated with 16 body joints. We adopt the standard train/val split in [1] to build the MPII training set and validation set, and conduct all the experiments on this dataset via first training the model on the MPII training set, and then evaluating it on the MPII validation set. Following [26], we use the head-normalized probability of correct keypoint (PCKh) [1] score as the evaluation metric on this dataset and report the PCKh@0.5 score.

**Implementation details.** On the MPII dataset, we also apply our method to various methods as our baselines, including Simple Baseline [33] and HRNet [26], with their respective backbones including ResNet-152, HRNet-W32, and HRNet-W48. We follow the original learning and optimization configurations for model training for both Simple Baseline [33] and HRNet [26]. Besides, same as the experiments on the COCO dataset, we also set the Sinkhorn entropic regularization parameter to 1 and the number of Sinkhorn iterations to 1000 in our experiments on the MPII dataset.

**Results on the MPII validation set.** In Tab. 3, we report the results on the MPII validation set. As shown, applying our proposed method on various baselines results in a consistent performance improvement, which demonstrates the effectiveness of our proposed method.

## 5.3 Ablation Studies

We conduct ablation studies on the COCO validation set via applying our proposed method on HRNet-W48 [26].

**Impact of involving the idea of sub-pixels in formulating $D$.** In our proposed method, we formulate the demanders $D$ by involving the idea of sub-pixel resolution. To investigate the impact of formulating the demanders $D$ in such a way, we compare our proposed method (**sub-pixel demanders formulation**) with a variant (**naive demanders formulation**). This variant still formulates the suppliers $S$ and the cost function $C$ in the same way, but formulates the demanders $D$ naively as a single demander at the center of the pixel containing the dot annotation, as shown in Fig. 2(a).

Table 3: The improvement of AP on MPII validation set when our proposed method is applied to various baselines.

| Method | Venue | Backbone | Input size | Mean | Hea | Sho | Elb | Wri | Hip | Kne | Ank |
|---|---|---|---|---|---|---|---|---|---|---|---|
| Integral Pose Regression[27] | ECCV 2018 | ResNet-101 | $256 \times 256$ | 87.9 | - | - | - | - | - | - | - |
| UDP[10] | CVPR 2020 | HRNet-W32 | $256 \times 256$ | 90.4 | 97.4 | 96.0 | 91.0 | 86.5 | 89.1 | 86.6 | 83.3 |
| DarkPose[39] | CVPR 2020 | HRNet-W32 | $256 \times 256$ | 90.6 | 97.2 | 95.9 | 91.2 | 86.7 | 89.7 | 86.7 | 84.0 |
| TokenPose[15] | ICCV 2021 | TokenPose-L/D6 | $256 \times 256$ | 90.1 | 97.1 | 95.9 | 91.0 | 85.8 | 89.5 | 86.1 | 82.7 |
| TokenPose[15] | ICCV 2021 | TokenPose-L/D12 | $256 \times 256$ | 90.1 | 97.2 | 95.8 | 90.7 | 85.9 | 89.2 | 86.2 | 82.3 |
| TokenPose[15] | ICCV 2021 | TokenPose-L/D24 | $256 \times 256$ | 90.2 | 97.1 | 95.9 | 90.4 | 86.0 | 89.3 | 87.1 | 82.5 |
| Removing Bias[7] | ICCV 2021 | ResNet-152 | $256 \times 256$ | 89.9 | - | - | - | - | - | - | - |
| Removing Bias[7] | ICCV 2021 | HRNet-W32 | $256 \times 256$ | 90.6 | - | - | - | - | - | - | - |
| Simple Baseline[33] | ECCV 2018 | ResNet-152 | $256 \times 256$ | 89.6 | **97.0** | 95.9 | 90.0 | 85.0 | **89.2** | 85.3 | 81.3 |
| **+ Ours** | | ResNet-152 | $256 \times 256$ | **90.3(↑0.7)** | **97.0** | **96.1** | **90.6** | **86.1** | **89.2** | **86.7** | **83.1** |
| HRNet[26] | CVPR 2019 | HRNet-W32 | $256 \times 256$ | 90.4 | 97.1 | 95.9 | 90.7 | 86.1 | 89.4 | 86.9 | 83.2 |
| **+ Ours** | | HRNet-W32 | $256 \times 256$ | **90.9(↑0.5)** | **97.3** | **96.2** | **91.2** | **86.8** | **90.1** | **87.4** | **84.1** |
| HRNet[26] | CVPR 2019 | HRNet-W48 | $256 \times 256$ | 90.5 | 96.9 | 96.0 | 90.9 | 86.2 | 89.6 | 87.1 | 83.5 |
| **+ Ours** | | HRNet-W48 | $256 \times 256$ | **90.9(↑0.4)** | **97.1** | **96.3** | **91.2** | **87.0** | **90.2** | **87.5** | **84.2** |

Table 4: Evaluation on the effectiveness of formulating the demanders $D$ involving the idea of sub-pixel resolution.

| Method | AP | AP$^{50}$ | AP$^{75}$ | AP$^M$ | AP$^L$ | AR |
|---|---|---|---|---|---|---|
| Baseline(HRNet-W48) | 77.1 | 91.8 | 83.8 | 73.5 | 83.5 | 81.8 |
| **Naive demanders formulation** | 77.9 | 92.5 | 84.8 | 74.5 | 83.9 | 82.4 |
| **Sub-pixel demanders formulation** | 78.8 | 92.5 | 85.1 | 75.0 | 85.3 | 83.1 |

Table 5: Evaluation on the number of Sinkhorn iterations.

| Method | AP | AP$^{50}$ | AP$^{75}$ | AP$^M$ | AP$^L$ | AR |
|---|---|---|---|---|---|---|
| Baseline(HRNet-W48) | 77.1 | 91.8 | 83.8 | 73.5 | 83.5 | 81.8 |
| 500 Sinkhorn iterations | 78.3 | 92.4 | 84.9 | 74.8 | 84.4 | 82.7 |
| 1000 Sinkhorn iterations | 78.8 | 92.5 | 85.1 | 75.0 | 85.3 | 83.1 |
| 1500 Sinkhorn iterations | 78.7 | 92.4 | 85.2 | 74.8 | 85.4 | 83.0 |

As shown in Tab. 4, our proposed method consistently outperforms this variant, which shows effectiveness of our **sub-pixel demanders formulation**.

**Impact of the number of Sinkhorn iterations.** For measuring the Earth Mover's Distance utilizing the Sinkhorn algorithm, we need to set the number of Sinkhorn iterations, which we set to 1000 in our experiments. We evaluate other choices of the number of Sinkhorn iterations in Tab. 5. As shown, all variants outperform the baseline method, and after the number of Sinkhorn iterations becomes larger than 1000, the model performance becomes stabilized. Hence, we set the number of Sinkhorn iterations to be 1000 in all our experiments.

**Qualitative results.** Some qualitative results are shown in Fig. 3. As shown, via formulating the cost function as the L2 distance, our proposed method can aggregate the pixel values in the predicted heatmap towards the dot annotation, and thus localize

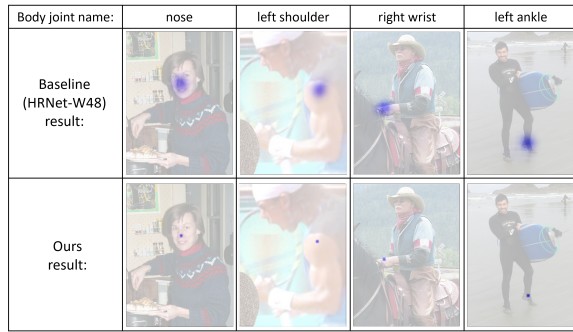

Figure 3: Qualitative results of our method formulating the optimization of the heatmap prediction as a distribution matching problem and the baseline method [26] using the Gaussian-smoothed heatmap as the optimization objective. As shown, our method localizes body joints much more compactly.

body joints much more compactly than the baseline method [26] relying on the Gaussian-smoothed heatmap. This demonstrates that our method can effectively optimize the model in the direction of accurately localizing the body joint.

## 6 Conclusion

In this paper, from a novel perspective, we formulate the optimization of the heatmap prediction as a distribution matching problem between the predicted heatmap and the dot annotation via calculating their Earth Mover's Distance. Our proposed method is simple yet effective, and can be easily applied to various 2D human pose estimation models. Our method achieves superior performance on the COCO dataset and the MPII dataset.

## Acknowledgments and Disclosure of Funding

This project is supported by the Ministry of Education, Singapore, under the SUTD Kickstarter Initiative Project (SKI 2021_02_06), National Research Foundation Singapore under its AI Singapore Programme (AISG-100E-2020-065), and SUTD Startup Research Grant.

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
