# Heatmap Distribution Matching for Human Pose Estimation (Supplementary)

**Haoxuan Qu**
SUTD
Singapore
haoxuan_qu@mymail.sutd.edu.sg

**Li Xu**
SUTD
Singapore
li_xu@mymail.sutd.edu.sg

**Yujun Cai**
NTU
Singapore
yujun001@e.ntu.edu.sg

**Lin Geng Foo**
SUTD
Singapore
lingeng_foo@mymail.sutd.edu.sg

**Jun Liu** *
SUTD
Singapore
jun_liu@sutd.edu.sg

## A    Additional Qualitative Results

In Fig. 1, we compare our method with the baseline method [3], and present qualitative results on the MPII validation set. Note that for the baseline method [3], we calculate the predicted coordinates from the predicted heatmap following its own design, and we calculate the predicted coordinates for our method following the process we discuss in Sec. 3.4 in the main paper. As shown, via formulating the cost function as the L2 distance, our method can aggregate the pixel values in the predicted heatmap towards the dot annotation, and thus output much more compact predicted heatmaps and localize body joints more accurately.

## B    Additional Ablation Studies

**Relation between the loss value and the body joint localization performance.** In the main paper, we mention that when minimizing the pixel-wise loss between the predicted heatmap and the GT Gaussian-smoothed heatmap, the model performance of body joint localization cannot be guaranteed to improve consistently. Here, we further conduct an experiment on the MPII validation set to demonstrate this. Specifically, we plot two pairs of curves from epoch 2 to epoch 15 of the model optimization process respectively, between the pixel-wise MSE loss on the MPII validation set and the model performance on the MPII validation set in Fig. 2(a), and between our proposed loss on the MPII validation set and the model performance on the MPII validation set in Fig. 2(b). As shown in Fig. 2(a), during the model optimization process, while the pixel-wise MSE loss measured on the MPII validation set decreases, the model performance of body joint localization measured on the same set (the MPII validation set) does not improve consistently. This means that the pixel-wise loss has limitations as the decrease of the pixel-wise loss is not directly relevant to the increase of the model performance. Compared with this, as shown in Fig. 2(b), as our proposed loss can aggregate the pixel values in the predicted heatmap directly towards the dot annotation, while our proposed loss measured on the MPII validation set decreases, a consistent model performance improvement on the

---

*Corresponding Author

36th Conference on Neural Information Processing Systems (NeurIPS 2022).

| Body joint name: | upper neck | left wrist | left knee | right knee |
|---|---|---|---|---|
| Baseline (HRNet-W48) result (predicted heatmaps & coordinates): | (764.4, 315.3) | (335.0, 77.6) | (780.7, 331.3) | (651.8, 889.1) |
| Ours result (predicted heatmaps & coordinates): | (755.9, 313.7) | (335.8, 76.1) | (777.2, 335.9) | (655.7, 877.5) |
| Ground truth coordinates: | (750.5, 297.7) | (337.0, 74.0) | (777.0, 342.0) | (658.0, 865.0) |

Figure 1: Qualitative results of our method and the baseline method [3].

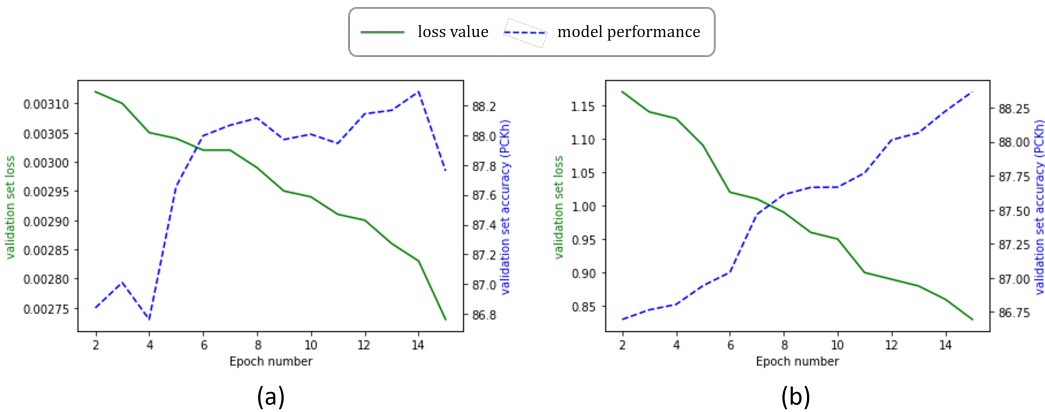

(a)                    (b)

Figure 2: Illustration of (a) the relation between the value of the pixel-wise MSE loss and the model performance of body joint localization, and (b) the relation between the value of our proposed loss and the model performance of body joint localization. As shown, when the value of our proposed loss becomes smaller, a consistent model performance improvement is achieved.

same set (the MPII validation set) is achieved. This demonstrates the effectiveness of our method as the decrease of our proposed loss is more directly relevant to the increase of the model performance.

**Comparison between our method and the method using the dot-annotated heatmap.** In our proposed method, we formulate the optimization of the heatmap prediction as a distribution matching problem between the predicted heatmap and the dot annotation. Here, to demonstrate the effectiveness of our method, besides comparing our method with the baseline method [3] (**Gaussian-smoothed heatmap**) which uses the Gaussian-smoothed heatmap as the optimization objective, we also compare our method with another baseline method (**dot-annotated heatmap**) that uses the same backbone as our method but uses the dot-annotated heatmap as the optimization objective and uses the pixel-wise MSE loss as the loss function. As shown in Tab. 1, our proposed method consistently outperforms both types of baseline methods, which shows the effectiveness of our method.

Table 1: Comparison between our method and the two types of baseline methods.

| Method | AP | AP$^{50}$ | AP$^{75}$ | AP$^M$ | AP$^L$ | AR |
|---|---|---|---|---|---|---|
| **Gaussian-smoothed heatmap** | 77.1 | 91.8 | 83.8 | 73.5 | 83.5 | 81.8 |
| **dot-annotated heatmap** | 46.6 | 86.0 | 45.1 | 42.9 | 52.9 | 58.5 |
| **Our method** | 78.8 | 92.5 | 85.1 | 75.0 | 85.3 | 83.1 |

## C   Additional Training Details

Remind that in the main paper, we formulate the optimization of the heatmap prediction as a distribution matching problem and construct a loss function following such a formulation as:

$$L_{Matching} = \sum_{k=1}^{K} L_k, \text{ where } L_k = E_{C^k}^{reg}(S^k, D^k) \tag{1}$$

where $S^k$, $D^k$, and $C^k$ respectively denote the constructed suppliers, demanders, and cost function for the $k$-th body joint. In this section, we introduce how we calculate the gradient of our proposed loss function. Remind that we derive the suppliers by first passing the predicted heatmap through a $relu$ activation function and then normalizing it. As the $relu$ activation function and the normalization process are differentiable, it is easy to calculate the gradient of $S^k$ w.r.t. the corresponding predicted heatmap. Below, we introduce how we calculate the gradient of $L_{Matching}$ w.r.t. $S^k$. Specifically, the Earth Mover's Distance $E_{C^k}^{reg}(S^k, D^k)$ can be written in its dual form as:

$$E_{C^k}^{reg}(S^k, D^k) = \max_{a^k \in R^{HW} \ \& \ b^k \in R^4} \quad \langle a^k, S^k \rangle + \langle b^k, D^k \rangle$$

$$\text{s.t.} \quad \forall n \in \{1, ..., H \times W\} \ \& \ m \in \{1, 2, 3, 4\}, a_n^k + b_m^k \leq C_{n,m}^k \tag{2}$$

Via writing the $E_{C^k}^{reg}(S^k, D^k)$ in its dual form as above, we can calculate the gradient of $L_{Matching}$ w.r.t. $S^k$ as:

$$\frac{\partial L_{Matching}}{\partial S^k} = \frac{\partial L_k}{\partial S^k} = \frac{\partial E_{C^k}^{reg}(S^k, D^k)}{\partial S^k} = a^k \tag{3}$$

Via calculating the gradient of $L_{Matching}$ w.r.t. $S^k$ and the gradient of $S^k$ w.r.t. its corresponding predicted heatmap respectively as discussed above, we can back propagate our proposed loss through the suppliers and the predicted heatmaps into the backbone model and update the model parameters correspondingly.

## D   Proof of Theorem 1 in the Main Paper

To prove Theorem 1 in the main paper, we first introduce some notations as follows. Note that all the notations we introduce here are the same as the notations we introduce in the main paper.

We denote $K$ the number of joints per input image $I$. Then we denote $\mathbf{H_{dot}} = \{H_{dot}^1, ..., H_{dot}^K\}$, $\mathbf{H_{Gau}} = \{H_{Gau}^1, ..., H_{Gau}^K\}$, and $\mathbf{H_{pred}} = \{H_{pred}^1, ..., H_{pred}^K\}$ respectively the corresponding $K$ GT dot-annotated heatmaps, GT Gaussian-smoothed heatmaps, and predicted heatmaps of the input image $I$. We denote $\mathcal{D}_{dot} = \{(I, \mathbf{H_{dot}})\}$ the joint distribution of the input image and the corresponding $K$ dot-annotated heatmaps, and $\mathcal{D}_{Gau} = \{(I, \mathbf{H_{Gau}})\}$ the joint distribution of the input image and the corresponding $K$ Gaussian-smoothed heatmaps. Besides, we denote $l_{MSE}(a, b) = \|a - b\|_2^2$ the pixel-wise MSE loss, and $\phi$ the model parameters where $\phi(I) = \mathbf{H_{pred}}$. After that, we denote $R(\mathcal{D}_{dot}, \phi, l_{MSE}) = \mathbb{E}_{(I, \mathbf{H_{dot}}) \sim \mathcal{D}_{dot}}[l_{MSE}(\phi(I), \mathbf{H_{dot}})]$ as the expected risk calculated between the predicted heatmaps and the GT dot-annotated heatmaps, and $R(\mathcal{D}_{Gau}, \phi, l_{MSE}) = \mathbb{E}_{(I, \mathbf{H_{Gau}}) \sim \mathcal{D}_{Gau}}[l_{MSE}(\phi(I), \mathbf{H_{Gau}})]$ as the expected risk calculated between the predicted heatmaps and the Gaussian-smoothed heatmaps.

**Theorem 1.** *The relationship between $R(\mathcal{D}_{dot}, \phi, l_{MSE})$ and $R(\mathcal{D}_{Gau}, \phi, l_{MSE})$ can be written as*

$$R(\mathcal{D}_{dot}, \phi, l_{MSE}) = R(\mathcal{D}_{Gau}, \phi, l_{MSE}) + 2 \times \mathbb{E}_{(I, \mathbf{H_{Gau}}) \sim \mathcal{D}_{Gau}}[\langle \mathbf{H_{pred}}, \mathbf{H_{Gau}} \rangle]$$

$$- 2 \times \mathbb{E}_{(I, \mathbf{H_{dot}}) \sim \mathcal{D}_{dot}}[\langle \mathbf{H_{pred}}, \mathbf{H_{dot}} \rangle] - C \tag{4}$$

*where $C = \mathbb{E}_{(I, \mathbf{H_{dot}}) \sim \mathcal{D}_{dot}}[\|\mathbf{H_{Gau}} - \mathbf{H_{dot}}\|_2^2]$ is a constant.*

*Proof.* Remind that both $\mathbf{H_{dot}}$ and $\mathbf{H_{Gau}}$ have the pixel representing the dot annotation to be one, and $\mathbf{H_{dot}}$ further has all the other pixels to be zero. Hence, the following equation holds:

$$\|\mathbf{H_{dot}}\|_2^2 = \langle \mathbf{H_{Gau}}, \mathbf{H_{dot}} \rangle \tag{5}$$

Hence, Theorem 1 can be proved as:

$$
\begin{aligned}
R(\mathcal{D}_{dot}, \phi, l_{MSE}) &= \mathbb{E}_{(I,\mathbf{H_{dot}})\sim\mathcal{D}_{dot}}[\|\mathbf{H_{dot}} - \mathbf{H_{pred}}\|_2^2] \\
&= \mathbb{E}_{(I,\mathbf{H_{dot}})\sim\mathcal{D}_{dot}}[\|\mathbf{H_{dot}}\|_2^2 + \|\mathbf{H_{pred}}\|_2^2 - 2 \times \langle \mathbf{H_{dot}}, \mathbf{H_{pred}} \rangle] \\
&= \mathbb{E}_{(I,\mathbf{H_{dot}})\sim\mathcal{D}_{dot}}[2 \times \|\mathbf{H_{dot}}\|_2^2 - \|\mathbf{H_{dot}}\|_2^2 + \|\mathbf{H_{pred}}\|_2^2 - 2 \times \langle \mathbf{H_{dot}}, \mathbf{H_{pred}} \rangle \\
&\qquad + 2 \times \langle \mathbf{H_{Gau}}, \mathbf{H_{pred}} \rangle - 2 \times \langle \mathbf{H_{Gau}}, \mathbf{H_{pred}} \rangle \\
&\qquad + \|\mathbf{H_{Gau}}\|_2^2 - \|\mathbf{H_{Gau}}\|_2^2] \\
&= \mathbb{E}_{(I,\mathbf{H_{dot}})\sim\mathcal{D}_{dot}}[\|\mathbf{H_{Gau}}\|_2^2 + \|\mathbf{H_{pred}}\|_2^2 - 2 \times \langle \mathbf{H_{Gau}}, \mathbf{H_{pred}} \rangle \\
&\qquad - \|\mathbf{H_{dot}}\|_2^2 - \|\mathbf{H_{Gau}}\|_2^2 + 2 \times \|\mathbf{H_{dot}}\|_2^2 \\
&\qquad + 2 \times \langle \mathbf{H_{Gau}}, \mathbf{H_{pred}} \rangle - 2 \times \langle \mathbf{H_{dot}} \cdot \mathbf{H_{pred}} \rangle] \\
&= \mathbb{E}_{(I,\mathbf{H_{dot}})\sim\mathcal{D}_{dot}}[\|\mathbf{H_{Gau}}\|_2^2 + \|\mathbf{H_{pred}}\|_2^2 - 2 \times \langle \mathbf{H_{Gau}}, \mathbf{H_{pred}} \rangle \\
&\qquad - \|\mathbf{H_{Gau}}\|_2^2 - \|\mathbf{H_{dot}}\|_2^2 + 2 \times \langle \mathbf{H_{Gau}}, \mathbf{H_{dot}} \rangle \\
&\qquad + 2 \times \langle \mathbf{H_{Gau}}, \mathbf{H_{pred}} \rangle - 2 \times \langle \mathbf{H_{dot}}, \mathbf{H_{pred}} \rangle] \\
&= \mathbb{E}_{(I,\mathbf{H_{dot}})\sim\mathcal{D}_{dot}}[\|\mathbf{H_{Gau}} - \mathbf{H_{pred}}\|_2^2 - \|\mathbf{H_{Gau}} - \mathbf{H_{dot}}\|_2^2 \\
&\qquad + 2 \times \langle \mathbf{H_{Gau}}, \mathbf{H_{pred}} \rangle - 2 \times \langle \mathbf{H_{dot}}, \mathbf{H_{pred}} \rangle] \\
&= R(\mathcal{D}_{Gau}, \phi, l_{MSE}) + 2 \times \mathbb{E}_{(I,\mathbf{H_{Gau}})\sim\mathcal{D}_{Gau}}[\langle \mathbf{H_{pred}}, \mathbf{H_{Gau}} \rangle] \\
&\qquad - 2 \times \mathbb{E}_{(I,\mathbf{H_{dot}})\sim\mathcal{D}_{dot}}[\langle \mathbf{H_{pred}}, \mathbf{H_{dot}} \rangle] - \mathbb{E}_{(I,\mathbf{H_{dot}})\sim\mathcal{D}_{dot}}[\|\mathbf{H_{Gau}} - \mathbf{H_{dot}}\|_2^2]
\end{aligned}
\tag{6}
$$

The last equality holds since $R(\mathcal{D}_{Gau}, \phi, l_{MSE}) = \mathbb{E}_{(I,\mathbf{H_{Gau}})\sim\mathcal{D}_{Gau}}[\|\mathbf{H_{Gau}} - \mathbf{H_{pred}}\|_2^2] = \mathbb{E}_{(I,\mathbf{H_{dot}})\sim\mathcal{D}_{dot}}[\|\mathbf{H_{Gau}} - \mathbf{H_{pred}}\|_2^2]$ and $\mathbb{E}_{(I,\mathbf{H_{Gau}})\sim\mathcal{D}_{Gau}}[\langle \mathbf{H_{pred}}, \mathbf{H_{Gau}} \rangle] = \mathbb{E}_{(I,\mathbf{H_{dot}})\sim\mathcal{D}_{dot}}[\langle \mathbf{H_{pred}}, \mathbf{H_{Gau}} \rangle]$. These are because the expectation calculated over $\mathcal{D}_{dot}$ and the expectation calculated over $\mathcal{D}_{Gau}$ are the same, as the element of the distribution $\mathcal{D}_{dot}$ (i.e., $(I, \mathbf{H_{dot}})$) and the element of the distribution $\mathcal{D}_{Gau}$ (i.e., $(I, \mathbf{H_{Gau}})$) are one to one correspondent to each other. $\square$

# E    Failure Cases

In Fig. 3, we show some failure cases of our method. As shown, in some extremely challenging cases (e.g., body joints under severe occlusion), both the baseline method and our method may not localize the body joints very accurately. This is also a limitation, and an important research problem in the task of pose estimation that we will take as a future direction.

# F    Dataset Licenses

We use the COCO dataset [2] by following the Creative Commons Attribution 4.0 License, and use the MPII dataset [1] by following the Simplified BSD License.

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

| Body joint name: | right elbow | left knee |
|---|---|---|
| Baseline (HRNet-W48) result (predicted heatmaps & coordinates): |  (240.1, 209.5) |  (543.3, 550.0) |
| Ours result (predicted heatmaps & coordinates): |  (233.1, 213.7) |  (549.4, 552.9) |
| Ground truth result (dot-annotated heatmaps & GT coordinates): |  (192.0, 217.0) |  (554.0, 600.0) |

Figure 3: Failure cases of joints under severe occlusion

[3] Ke Sun, Bin Xiao, Dong Liu, and Jingdong Wang. Deep high-resolution representation learning for human pose estimation. In *Proceedings of the IEEE/CVF Conference on Computer Vision and Pattern Recognition*, pages 5693–5703, 2019.