# OpenReview forum: "Heatmap Distribution Matching for Human Pose Estimation"
_NeurIPS.cc/2022/Conference — NeurIPS 2022 Accept_

### Official Review · Reviewer_ggaN · 2022-06-27

**Rating:** 7
**Confidence:** 5
**Soundness:** 3 good
**Presentation:** 3 good
**Contribution:** 3 good

**Summary:**

This paper proposes using EMD to measure the difference between the GT and the predicted heatmaps. Compared with L2 distance, EMD avoids the choice of standard deviation and is more consistent with the final objective of pose estimation. The proposed loss function can largely improve the performance of pose-estimation methods.

**Questions:**

See the questions above.

**Ethics Review Area:**

["I don’t know"]

**Limitations:**

The idea is elegant and effective. My only concern is about the training time. However, given its marvelous performance, a longer training time is also acceptable to some degree.

**Strengths And Weaknesses:**

Strengths:
The paper is well written and easy to follow. The proposed method is simple yet can largely improve the performance of pose-estimation methods.

Questions:
1. According to Line144, the authors seem to use Sinkhorn algorithm to solve the EMD. Is this process differentiable? Could you please supplement more details about how the gradients are passed backward?

2. According to Eq 5, the EMD is respectively calculated for each keypoint.  How long does it take to solve the EMD for one keypoint? And it seems that the calculation of EMD also cannot be paralleled among a mini-batch. Will the proposed method largely extend the training time?

3. It seems that EMD can also be applied to the heatmaps with multiple keypoints, which means that the proposed functions can also be applied for muli-human pose estimation. Have the author tried this?

---

> ### Author Response · Authors · 2022-08-02
> **Response to reviewer ggaN's review (1/2)**
>
> We thank the reviewer for the thoughtful comments. In the following, we seek to address each of the concerns.
>
> >**Q1:** *"According to Line144, the authors seem to use Sinkhorn algorithm to solve the EMD. Is this process differentiable? Could you please supplement more details about how the gradients are passed backward?"*
>
> **A1:** This process is differentiable. The analysis of (1) how the gradients are calculated from the Sinkhorn algorithm and (2) how the gradients are passed backward from our proposed loss $L_{Matching}$ into the backbone model was shown in Section C of our Supplementary. Below we also show that analysis:
>
> **(1) How the gradients are calculated from the Sinkhorn algorithm.**
>
> To calculate the gradients from the Sinkhorn algorithm, for the $k$-th body joint, we first rewrite the corresponding Earth Mover's Distance $E^{reg}_{C^k}(S^k, D^k)$ in its dual form (as shown in Eq. 2 of our Supplementary).
>
> Then from its dual form, we can calculate the gradient of $E^{reg}_{C^k}(S^k, D^k)$ w.r.t. the supplier $S^k$ as $a^k$. Note that $a^k$ is already provided during computing the Earth Mover's Distance using Sinkhorn algorithm so calculating $a^k$ does not introduce additional computational cost.
>
> **(2) How the gradients are passed backward from $L_{Matching}$ into the backbone model.**
>
> To pass the gradients backward from $L_{Matching}$ into the backbone model, for the $k$-th body joint, we first calculate the gradient of $L_{Matching}$ w.r.t. $S^k$ as $\frac{\partial L_{Matching}}{\partial S^k} = \frac{\partial L_k}{\partial S^k} = \frac{\partial E^{reg}_{C^k}(S^k, D^k)}{\partial S^k} = a^k$. Then we calculate the gradient of $S^k$ w.r.t. its corresponding predicted heatmap, which is clearly differentiable as the supplier is formulated from the predicted heatmap through $relu$ and normalization operations (as shown in Eq. 3 of our paper), and both $relu$ and normalization operations are differentiable. This means that the process is differentiable.
>
> More details about how the gradients are passed backward are shown in our Supplementary. Kindly refer to Section C of our Supplementary for more details.
>
>
> >**Q2:** *"According to Eq 5, the EMD is respectively calculated for each keypoint. How long does it take to solve the EMD for one keypoint? And it seems that the calculation of EMD also cannot be paralleled among a mini-batch. Will the proposed method largely extend the training time?"*
>
> **A2:** (1) In the total training time of a mini-batch of data, solving the EMD takes roughly **33%** of the time. Also note that the calculations of EMD is only conducted during training, and thus the inference time with and without our proposed method are almost the same.
>
> (2) As the calculations of EMD of each keypoint are independent to each other, these calculations can be paralleled among a mini-batch, and thus are efficient.
>
> (3) Below we show the training time of the original HRNet-W48 [26] without the calculation of EMD and the proposed method (HRNet-W48 + Ours) with the calculation of EMD on COCO dataset on RTX 3090 GPU. For fair comparison, in the experiment below, we train both the original HRNet-W48 [26] and the proposed method from scratch.
>
> | Method  | Training Time | Inference Time | Performance(AP) |
> |---|---|---|---|
> | **HRNet-W48 [26]**  | 2 days | 0.02 second per sample | 77.1 |
> | **HRNet-W48 + Ours**  | 3 days | 0.02 second per sample | 78.8 |
>
> As can be seen in the table above, though the proposed method achieves obviously better performance, it only brings relatively small increase of the training time. Besides, as our proposed method does not need to calculate EMD during inference, the inference time with and without our proposed method are almost the same.

---

> > ### Author Response · Authors · 2022-08-02
> > **Response to reviewer ggaN's review (2/2)**
> >
> > >**Q3:** *"It seems that EMD can also be applied to the heatmaps with multiple keypoints, which means that the proposed functions can also be applied for multi-human pose estimation. Have the author tried this?"*
> >
> > **A3:** We also apply our method to the heatmaps with multiple keypoints. Specifically, we apply our method on two commonly used backbone models where the outputs are heatmaps with multiple keypoints (i.e., HrHRnet-W32 [c] and HrHRnet-W48 [c]). As shown below, on both backbones, our method achieves consistent performance improvement and achieves SOTA performance on both multi-human COCO validation set and multi-human COCO test-dev set.
> >
> > | Method  | Set | Input Size | $AP$ | $AP^{50}$  | $AP^{75}$  | $AP^{M}$  | $AP^{L}$ |
> > |---|---|---|---|---|---|---|---|
> > | HrHRnet-W32 [c] | multi-human COCO validation | 512 | 69.9 | 87.1 | 76.0 | 65.3 | 77.0 |
> > | HrHRnet-W32 + Ours | multi-human COCO validation | 512 | 71.6 | 88.7 | 77.7 | 66.7 | 78.5 |
> >
> > | Method  | Set | Input Size | $AP$ | $AP^{50}$  | $AP^{75}$  | $AP^{M}$  | $AP^{L}$ |
> > |---|---|---|---|---|---|---|---|
> > | HrHRnet-W32 [c] | multi-human COCO test-dev | 512 | 69.0 | 89.0 | 75.8 | 64.4 | 75.2 |
> > | HrHRnet-W32 + Ours | multi-human COCO test-dev | 512 | 70.7 | 90.0 | 77.7 | 66.3 | 76.8 |
> >
> > | Method  | Set | Input Size | $AP$ | $AP^{50}$  | $AP^{75}$  | $AP^{M}$  | $AP^{L}$ |
> > |---|---|---|---|---|---|---|---|
> > | HrHRnet-W48 [c] | multi-human COCO validation | 640 | 72.1 | 88.4 | 78.2 | 67.8 | 78.3 |
> > | HrHRnet-W48 + Ours | multi-human COCO validation | 640 | 73.5 | 89.9 | 79.4 | 69.7 | 79.1 |
> >
> > | Method  | Set | Input Size | $AP$ | $AP^{50}$  | $AP^{75}$  | $AP^{M}$  | $AP^{L}$ |
> > |---|---|---|---|---|---|---|---|
> > | HrHRnet-W48 [c] | multi-human COCO test-dev | 640 | 70.5 | 89.3 | 77.2 | 66.6 | 75.8 |
> > | HrHRnet-W48 + Ours | multi-human COCO test-dev | 640 | 72.2 | 90.9 | 79.0 | 68.3 | 77.4 |
> >
> >
> > [c] Cheng, Bowen, et al. HigherHRNet: Scale-aware representation learning for bottom-up human pose estimation. CVPR, 2020.

---

### Official Review · Reviewer_aExd · 2022-07-10

**Rating:** 6
**Confidence:** 3
**Soundness:** 3 good
**Presentation:** 3 good
**Contribution:** 3 good

**Summary:**

1. The paper analyzes that the commonly used minimizing pixel-wise loss cannot guarantee that the performance of pose estimation increase.
2. The paper proposes using Earth Mover’s Distance to match the distributions between GT dot heatmap and predicted heatmap. The experiment shows that this loss can improve the performance by 0.9-1.8% AP.


**Questions:**

1. Why baseline dot-annotated heatmap (MSE loss) doesn't work, but proposed method does. Is it because the MSE loss (gradient) is too sparse? What role does Heatmap normalization play here? The authors may be able to provide more thorough analysis on this problem.

2. The authors can compare in more detail with UDP and Darkpose, which fix some issues in native Gaussian Heatmap.

3. Compared with baseline, does the running time increase? This helps to assess the practicality of the method.


**Limitations:**

This paper does not provide limitations and potential negative societal impact. Please refer to the Questions section on limitations.

**Strengths And Weaknesses:**

**Strengths:**

1. The heatmap is a basic and important tool in human pose estimation. The paper analyzes the problem of commonly used pixel-wise heatmap loss, and formulate the optimization of the heatmap prediction as a distribution matching problem by Earth Mover’s Distance. It is novel and technically sound.

2. It seems that the proposed loss is model-agnostic that can be applied in most pose estimator and improve the performance.

**Weaknesses:**

The paper lacks the complexity analysis of the proposed method, such as the training and inference time.

---

> ### Author Response · Authors · 2022-08-02
> **Response to reviewer aExd's review (1/2)**
>
> We thank the reviewer for the thoughtful comments. In the following, we seek to address each of the concerns.
>
> >**Q1:** *"The paper lacks the complexity analysis of the proposed method, such as the training and inference time. Compared with baseline, does the running time increase? This helps to assess the practicality of the method."*
>
> **A1:** We show the training time of the baseline (HRNet-W48 [26]) and the proposed method (HRNet-W48 + Ours) below with the same backbone model on COCO dataset on RTX 3090 GPU. For fair comparison, in the experiment below, we train both the baseline and the proposed method from scratch.
>
> | Method  | Training Time | Inference Time | Performance(AP) |
> |---|---|---|---|
> | **Baseline (HRNet-W48 [26])**  | 2 days | 0.02 second per sample | 77.1 |
> | **HRNet-W48 + Ours**  | 3 days | 0.02 second per sample | 78.8 |
>
>
> As shown, though our method achieves obviously better performance, it only brings relatively small increase of the training time. Moreover, as the calculations of EMD is only conducted during training, the inference time with and without the proposed method are almost the same.
>
> >**Q2:** *"Why baseline dot-annotated heatmap (MSE loss) does not work, but proposed method does?"*
>
> **A2:** The baseline dot-annotated heatmap (MSE loss) does not work since it faces a hard training process and thus has a weak model performance, as shown by [28].
>
> Specifically, the GT dot-annotated heatmap has the same zero value for all pixels except the pixel representing the dot annotation of the body joint.
> Owing to such extreme sparsity, it is difficult to optimize the heatmap prediction by properly pushing the activated region on the predicted heatmap to move towards the GT position. More specifically, in heatmap estimation, we need to activate the pixel corresponding to the GT position, however, when the activated region in the predicted heatmap is not adjacent to the GT dot annotation, no matter which direction the activated region moves on the heatmap in an optimization step, the corresponding MSE loss could still keep the same. In other words, the baseline dot-annotated heatmap (MSE loss) can have difficulties in optimizing the heatmap prediction by moving (aggregating) the activation towards the dot annotation position. Hence, the baseline dot-annotated heatmap (MSE loss) faces a hard training process (as shown in [28]) and thus a weak model performance (e.g., more than **30%** of performance drop on COCO validation set, as shown in Tab. 1 of our Supplementary).
>
> In contrast, in our proposed method, we formulate the cost function between each pair of supplier and demander as their L2 distance (as shown in Line 207-209 of our paper). Because of this, when the pixel values (activated region) are moving (aggregating) towards the dot annotation position, the corresponding cost (loss) will decrease. Hence, minimizing the EMD based on such a cost function can explicitly move (aggregate) the pixel values in the predicted heatmap towards the dot annotation position. Therefore, when minimizing the proposed loss function, our method can help to achieve a more consistent model performance improvement.
>
> >**Q3:** *What role does Heatmap normalization play here?*
>
> **A3:** A prerequisite for being able to calculate the Earth Mover's Distance between the suppliers $S$ and the demanders $D$ is that the total units of mass stored by $S$ needs to be the same as the total units of mass required by $D$. Thus, the role Heatmap normalization plays here is to ensure that this prerequisite is met. Specifically, we use Heatmap normalization to ensure that the total units of mass stored by the suppliers $S$ is 1. Then as the total units of mass required by the demanders $D$ is also 1, the total units of mass stored by $S$ and the total units of mass required by $D$ can then be guaranteed to be the same.

---

> > ### Author Response · Authors · 2022-08-02
> > **Response to reviewer aExd's review (2/2)**
> >
> > >**Q4:** *"The authors can compare in more detail with UDP and Darkpose, which fix some issues in native Gaussian Heatmap."*
> >
> > **A4:** Thanks for your suggestion. As shown in Tables 1-3 of our paper, we have compared the performance of our method and these two methods (UDP and Darkpose) on both COCO and MPII datasets. Our method achieves better performance than these two methods, demonstrating the effectiveness of our method.
> >
> > Here we also explain the difference between our method and these two methods.
> > Specifically, while fixing some issues in Gaussian Heatmap, both UDP and Darkpose still use Gaussian Heatmap as the target Heatmap in the Heatmap optimization process: (1) UDP proposes to construct two offset maps in addition to the Gaussian heatmap to reduce the statistical error in standard encoding-decoding; (2) Darkpose proposes to place the center of the Gaussian kernel used to construct the Gaussian heatmap directly at the position of the GT coordinates, instead of the center of the nearest pixel, to reduce the quantisation error in the standard coordinate encoding process. In contrast, in our method, instead of using any Gaussian heatmap, we formulate the optimization of the heatmap as a mass transportation problem directly between the predicted heatmap and the dot annotation. We will add more detailed discussions to paper.

---

### Official Review · Reviewer_1LPW · 2022-07-11

**Rating:** 5
**Confidence:** 5
**Soundness:** 2 fair
**Presentation:** 2 fair
**Contribution:** 2 fair

**Summary:**

Pose estimation performance may not be consistently improved when optimizing the heatmap prediction by minimizing the mean-squared error(MSE). The paper proposes to utilize the Sinkhorn Distance as the optimizing target.

**Questions:**

Please refer to Weaknesses

**Strengths And Weaknesses:**

### Strengths

1. The paper points out and analyzes the misalignment between the training loss and the final performance.
1. Sinkhorn Distance is proposed to replace the MSE loss to achieve a more consistent performance improvement during training.
1. The proposed method is reported to have notable performance improvements on COCO and MPII.

### Weaknesses

#### Novelty & Contribution

1. Formulating the heatmap optimization as a distribution matching problem is not a novel perspective. Previous methods represent the GT coordinates in either Categorial or Gaussian distribution and optimize the predictions towards the GT distribution by minimizing MSE.
1. Proposing to replace MSE with Sinkhorn Distance does not have enough contribution.

#### Presentation

1. There are inconsistent presentations. In my perspective, the limitation to address lies in the pixel-wise loss (e.g. MSE), which is claimed in the Introduction(line45-52).
   + However, in Abstract(line4-7), it's claimed that using the Gaussian-smoothed heatmap as the optimization objective is the limitation of previous methods
   + The paper's target should not be 'bypassing the step of constructing the GT heatmap', but to alleviate or solve the limitation of loss.
   + Missing analysis on why the proposed method can alleviate or solve the limitation of loss.
2. It's misleading to claim bypassing the step of constructing the GT heatmap. The demander $D^k$ is just another type of GT heatmap and still needs the construction step. The naïve demander formulation is the same as the dot-annotated heatmap.

#### Experiments

1. Missing important reports and comparisons on the training time between the baseline and the proposed method. As the Sinkhorn Distance takes 1000 iterations and each iteration has calculations on $(H_{hm} * W_{hm}) \times (H_{hm} * W_{hm})$ dim matrixes, it's presumed to be time-consuming.
2. Missing comparisons between the MSE loss and the proposed loss with the same type of target heatmap

---

> ### Author Response · Authors · 2022-08-02
> **Response to reviewer 1LPW's review (1/3)**
>
> We thank the reviewer for the thoughtful comments. In the following, we seek to address each of the concerns.
>
> >**Q1: Novelty & Contribution** *"Formulating the heatmap optimization as a distribution matching problem is not a novel perspective. Previous methods represent the GT coordinates in either Categorical or Gaussian distribution and optimize the predictions towards the GT distribution by minimizing MSE."*
>
> **A1:** In human pose estimation, the heatmap-based methods are a popular category of methods, and *the heatmap is a basic and important tool* (as mentioned by Reviewer aExd).
>
> To optimize the heatmap prediction, the pioneering heatmap-based human pose estimation method (i.e., "Joint Training of a Convolutional Network and a Graphical Model for Human Pose Estimation", NIPS 2014) proposed a heatmap optimization pipeline to optimize the heatmap prediction via minimizing the MSE loss between the predicted heatmap and the Gaussian-smoothed heatmap. Currently, this pipeline has become the most **commonly used pipeline** in human pose estimation.
>
> As pointed out by Reviewer wVS3, in our paper, we *present an analysis of why minimizing the Gaussian heatmap with l2 loss* (i.e., the **commonly used pipeline**) *is not optimal* (i.e., by using this pipeline, a misalignment exists between the training loss and the final performance). After the analysis, to handle the problem, rather than measuring the MSE loss between the predicted heatmap and the Gaussian-smoothed heatmap (i.e., instead of using either the MSE loss or the Gaussian-smoothed heatmap), we propose a new pipeline that formulates the optimization of the heatmap as a mass transportation problem directly between the predicted heatmap and the dot annotation. By doing so, we can directly minimize the difference between the predicted heatmap and the GT dot annotation, and thus the misalignment problem between the loss function and the final performance can be tackled.
>
> Note that besides the aforementioned **commonly used pipeline**, in our paper, we also discuss another possible heatmap optimization pipeline - the **naive pipeline**. To optimize the heatmap prediction, the **naive pipeline** first constructs the dot-annotated heatmap from the GT coordinates and then minimizes the MSE loss between the predicted heatmap and the constructed dot-annotated heatmap. However, since the dot-annotated heatmap has the same zero value for all pixels except the pixel representing the dot annotation of the body joint, this naive pipeline can lead to a hard training process and a very significant performance drop (more than **30%** of performance drop on COCO validation set compared to the **commonly used pipeline**), as indicated in [28] and also demonstrated in Tab. 1 of our Supplementary. Hence, to the best of our knowledge, this naive pipeline is not used in any state-of-the-art human pose estimation works.
>
> The very significant performance drop of this **naive pipeline** also demonstrates the large difference between this pipeline and our proposed method, as our proposed method can *largely improve the performance of pose estimation methods* (as mentioned by Reviewer ggaN) on both COCO and MPII datasets.
>
> **In summary, heatmap is a basic tool for pose estimation, while we analyse that there is an important misalignment problem in the popularly-used heatmap techniques and propose a new method for addressing the problem.**
>
> **To the best of our knowledge, we are the first to formulate the optimization of heatmap prediction as a mass transportation problem, and thus can minimize the difference between the predicted heatmap and the dot annotation directly, which thus tackles the important misalignment problem in the commonly-used heatmap optimization pipeline, which is crucial since the heatmap is a basic and important tool in human pose estimation (as mentioned by Reviewer aExd). Therefore, we think that the contribution of our paper can be significant and meaningful to the community.**
>
> Note that in our paper, the term **distribution matching** actually refers to **mass transportation between the supplier and the demander**. We will make this clearer in the revised version.

---

> > ### Author Response · Authors · 2022-08-02
> > **Response to reviewer 1LPW's review (2/3)**
> >
> > >**Q2: Novelty & Contribution** *"Proposing to replace MSE with Sinkhorn Distance does not have enough contribution."*
> >
> > **A2:**
> > As mentioned in the above **A1**, since *heatmap is a basic and important tool in human pose estimation*, in our work, we aim to explore how to better optimize such an important heatmap prediction process.
> > Hence, the contributions of our work lie in: (1) We present an analysis that the commonly used heatmap optimization pipeline *is not optimal* and can result in a misalignment problem between the training loss and the final performance.
> > (2) We propose a novel solution to address the misalignment problem.
> >
> > In specific, to handle this problem, we propose a novel pipeline to formulate the optimization of the heatmap as a mass transportation problem directly between the predicted heatmap and the dot annotation. As a result, we can directly minimize the difference between the predicted heatmap and the dot annotation, and the misalignment problem between the loss function and the final performance can thus be tackled.
> > This implies that replacing MSE with Sinkhorn Distance is just a part of our proposed pipeline, and is only a part of our contributions.
> >
> > >**Q3: Presentation** *"There are inconsistent presentations. In my perspective, the limitation to address lies in the pixel-wise loss (e.g. MSE), which is claimed in the Introduction(line45-52). However, in Abstract(line4-7), it's claimed that using the Gaussian-smoothed heatmap as the optimization objective is the limitation of previous methods."*
> >
> > **A3:** As mentioned in the above **A1**, most heatmap-based pose estimation methods optimize the heatmap prediction through a commonly used pipeline via using the **pixel-wise loss** (e.g. MSE) as the loss function and using the **Gaussian-smoothed heatmap** as the optimization objective.
> > Thus, the limitation to be addressed is the misalignment problem in such a pipeline.
> > We will make this clearer in the revised version.
> >
> >
> > >**Q4: Presentation** *"The paper's target should not be 'bypassing the step of constructing the GT heatmap', but to alleviate or solve the limitation of loss. [...] The demander $D^k$ is just another type of GT heatmap and still needs the construction step. The naive demander formulation is the same as the dot-annotated heatmap."*
> >
> > **A4:** Thanks for your suggestion. We will rephrase 'bypassing the step of constructing the GT heatmap' as 'handling the misalignment problem in the commonly used pipeline' in the revised version.
> >
> > >**Q5: Presentation** *"Analysis on why the proposed method can alleviate or solve the limitation of loss."*
> >
> > **A5:**
> > Below, we analyze why the proposed method can handle the limitation of loss (i.e., the misalignment problem between the training loss and the final performance). Specifically, this is because we formulate the cost function between each pair of supplier and demander as their L2 distance (as shown in Line 207-209 of our paper). Therefore, when the pixel values are moved towards the GT dot annotation, the corresponding cost will decrease. Hence, minimizing the EMD based on such a cost function can explicitly aggregate the pixel values in the predicted heatmap towards the dot annotation. This also means that under such a cost function, we are directly minimizing the distance between the location of the predicted body joint and the location of the dot annotation. As a result, the training loss and the final pose estimation (body joint localization) performance are better aligned in our method.

---

> > > ### Author Response · Authors · 2022-08-02
> > > **Response to reviewer 1LPW's review (3/3)**
> > >
> > > >**Q6: Experiments** *"Comparisons on the training time between the baseline and the proposed method. As the Sinkhorn Distance takes 1000 iterations and each iteration has calculations on $(H_{hm} * W_{hm}) \times (H_{hm} * W_{hm})$ dim matrixes, it's presumed to be time-consuming."*
> > >
> > > **A6:** We show the training time of  the baseline (HRNet-W48 [26]) and the proposed method (HRNet-W48 + Ours) below with the same backbone model on COCO dataset on RTX 3090 GPU. For fair comparison, in the experiment below, we train both the baseline and the proposed method from scratch.
> > >
> > > | Method  | Training Time | Inference Time | Performance(AP) |
> > > |---|---|---|---|
> > > | **Baseline (HRNet-W48 [26])**  | 2 days | 0.02 second per sample | 77.1 |
> > > | **HRNet-W48 + Ours**  | 3 days | 0.02 second per sample | 78.8 |
> > >
> > > As shown, though our method achieves obviously better performance, it only brings relatively small increase of training time. Note that although the calculation of Sinkhorn Distance takes iterations, its computation cost is still not much, when compared to the computation cost of the whole backbone.
> > >
> > > Also note that the calculations of Sinkhorn Distance is only conducted during training. Thus the inference time of the baseline and our method are almost the same.
> > >
> > >
> > > >**Q7: Experiments** *"Missing comparisons between the MSE loss and the proposed loss with the same type of target heatmap."*
> > >
> > > **A7:** Below we compare our proposed pipeline with a variant that treats the sub-pixel demander as the target heatmap and optimizes the heatmap prediction via minimizing the MSE loss between the predicted heatmap and the sub-pixel demander. Note that both experiments are conducted with the same backbone model (HRNet-W48) and on the same set (COCO validation set). As shown below, our proposed pipeline outperforms the variant by a large margin under the same type of target heatmap, demonstrating the effectiveness of our proposed method which can consistently aggregate the pixel values in the predicted heatmap towards the dot annotation, and thus lead to a better model performance.
> > >
> > > | Method  | $AP$ | $AP^{50}$  | $AP^{75}$  | $AP^{M}$  | $AP^{L}$ | $AR$ |
> > > |---|---|---|---|---|---|---|
> > > | **Sub-pixel demander (MSE loss)**  | 50.7 | 86.8 | 49.0 | 46.4 | 55.2 | 61.1 |
> > > | **Ours**  | 78.8 | 92.5 | 85.1 | 75.0 | 85.3 | 83.1 |

---

> > > > ### Comment · Reviewer_1LPW · 2022-08-07
> > > > **Why not use the Gaussian heatmap as the demander?**
> > > >
> > > > 1. In **Q7: Experiments** *"Missing comparisons between the MSE loss and the proposed loss with the same type of target heatmap."*, I meant Missing comparisons with the same Gaussian heatmap. Why not use the Gaussian heatmap as the demander/target? A gaussian heatmap can also, even better, reduce the quantization error, which is the crucial reason for proposing a sub-pixels demander.
> > > >
> > > > 2. Besides, comparisons between sub-pixels demander and Gaussian heatmap are missing. Is a sub-pixels demander better than a gaussian heatmap? If not, the difference between the proposed "novel pipeline" and the previous "Gaussian heatmap and MSE" pipeline is only the existing Sinkhorn Distance loss.
> > > >
> > > > 3. In **A6**, I'm familiar with the training of HRNet-W48 on COCO. If trained on a single RTX 3090 GPU, it can not be done in three days. I suggest re-checking the experiment.

---

> > > > > ### Author Response · Authors · 2022-08-09
> > > > > **Response to "Why not use the Gaussian heatmap as the demander?"**
> > > > >
> > > > > We thank the reviewer for the additional thoughtful discussions. In the following, we seek to address each of the concerns.
> > > > >
> > > > > >**Q8:** *"In Q7: Experiments [...], I meant Missing comparisons with the same Gaussian heatmap."*
> > > > >
> > > > > **A8:** Below, we show the comparisons with the same Gaussian heatmap as the target. Note that all the experiments are conducted with the same backbone model (HRNet-W48) and on the same set (COCO validation set).
> > > > >
> > > > > | Method  | $AP$ | $AP^{50}$  | $AP^{75}$  | $AP^{M}$  | $AP^{L}$ | $AR$ |
> > > > > |---|---|---|---|---|---|---|
> > > > > | **Gaussian heatmap (MSE loss)**  | 77.1 | 91.8 | 83.8 | 73.5 | 83.5 | 81.8 |
> > > > > | **Gaussian heatmap (Sinkhorn Distance loss)**  | 77.7 | 92.2 | 84.3  | 74.0 | 83.8 | 82.3 |
> > > > >
> > > > > As shown, with the same Gaussian heatmap as the target, the proposed Sinkhorn Distance loss leads to a performance improvement compared to MSE loss.
> > > > >
> > > > > >**Q9:** *"Why not use the Gaussian heatmap as the demander/target? A Gaussian heatmap can also, even better, reduce the quantization error, which is the crucial reason for proposing a sub-pixels demander.""Besides, comparisons between sub-pixels demander and Gaussian heatmap are missing. Is a sub-pixels demander better than a Gaussian heatmap? If not, the difference between the proposed "novel pipeline" and the previous "Gaussian heatmap and MSE" pipeline is only the existing Sinkhorn Distance loss."*
> > > > >
> > > > > **A9:** Below we further explain why we do not use the Gaussian heatmap as the demander/target. (1) As shown in Line 41-45 of our paper and also in [17], by using the Gaussian heatmap as the demander/target, the standard deviations of the Gaussian distributions often need to be carefully chosen, which is non-trivial. (2) As shown in Fig. 3 and Line 341-347 of our paper, while the human pose estimation task aims to localize body joints accurately, by using the Gaussian heatmap as the demander/target, the predicted heatmap is not very compact. This can lead to difficulties in accurately localizing the body joints.
> > > > >
> > > > > Meanwhile, our proposed pipeline can alleviate the misalignment problem between the training loss and the final body joint localization performance (as elaborated in **A5**). Also, as shown in Fig. 3 of our paper, by using our proposed pipeline with sub-pixels demander, a more compact body joint localization can be achieved. Thus, the body joints can be localized more accurately. By using our proposed pipeline, we also bypass the step of choosing proper standard deviations.
> > > > >
> > > > > We also compare between sub-pixels demander and Gaussian heatmap with the same backbone model (HRNet-W48) and on the same set (COCO validation set). With the same Sinkhorn Distance loss, the variant using the sub-pixels demander further improves the performance over the variant using the Gaussian heatmap (78.8 vs 77.7 for AP), demonstrating the effectiveness of the sub-pixels demander in our proposed pipeline.
> > > > >
> > > > > >**Q10:** *"In A6, I'm familiar with the training of HRNet-W48 on COCO. If trained on a single RTX 3090 GPU, it can not be done in three days. I suggest re-checking the experiment."*
> > > > >
> > > > > **A10:** In many recent human pose estimation works [33, 26, 37], during training, the batch size is set to either 128 or 256, and multiple GPU cards are used. Following this setting, in our work, during training, we set the batch size to 256 and run the experiments using a GPU cluster with RTX 3090 GPU, which has 8 GPU cards in it. We will make this clearer in the revised version.

---

> > > > > > ### Comment · Reviewer_1LPW · 2022-08-10
> > > > > > **Thanks for your replies**
> > > > > >
> > > > > > Your replies generally answered my concerns and thus I change my rating. The suggestion of clarifying the core idea and supplementing the missing ablation study in the revised version, as mentioned in *Weakness*, still holds.

---

### Official Review · Reviewer_wVS3 · 2022-07-11

**Rating:** 6
**Confidence:** 5
**Soundness:** 3 good
**Presentation:** 3 good
**Contribution:** 2 fair

**Summary:**

This paper presents a method to estimate 2d locations of joints for human pose estimation task. Authors present a technique loss function to reduce the trade off between dot notation and heatwap notation to estimate the joint locations. For this purpose authors propose using earth mover's distance as the loss function. The paper proposes the construction of supplier and demander values as well as the cost function associated. The final minimization is obtained by using the sink horn's iteration method. Authors also present an analysis of why minimize the Gaussian heatmap with l2 loss is not optimal. Experiments on mpii and coco datasets are provided

**Questions:**

Point (2) from the weakness section. If the objective is to decrease the area of localization it is important to observe a naive approach first.
Several, paper have shown that to work for the cash of facial landmark localization.

Point (3) please provide failure cases in the experiment section. The paper does not talk about any limitations of the proposed approach
.

Point (4) Earth movers distance can be difficult to train from scratch. This limits the paper to only be applied to already existing pre trained networks.

**Limitations:**

The authors have not talked about any limitation of the paper. Conceptual or societal

Suggestions are already included is the previous sections

**Strengths And Weaknesses:**

Strengths
1. The paper is well rewritten and easy to understand with almost no typos
2. Although not very complicated paper the paper present an easy way to improve the quality of joint localization
3. The paper presents an analysis demonstrating the limitation of the methods.

Weakness
1. Several other paper have shown the limitation of heatmap regression and proposed solution to tackle those limitation. For ex, integral pose regression shows that it achieves similar numbers on mpii dataset, which the authors have not included in the table, however the paper does show companion with pose regression for coco dataset. Similarly (7) also shows similar performance on mpii dataset.

2. It will be interesting to observe how does this method compare to a heatmap regression based method where a multistep straining is applied where with each step of training the standard deviation is decreased

3. It will be good to see the failure cases of this method.

4. were the networks in the experiment initialized from already available pretrained weights? If that in the case it will be interesting to observe how the networks train when trained from scratch. Do the networks converge withEMD when trained from scratch.

---

> ### Author Response · Authors · 2022-08-02
> **Response to reviewer wVS3's review (1/2)**
>
> We thank the reviewer for the thoughtful comments. In the following, we seek to address each of the concerns.
>
>
> >**Q1:** *"Several other papers have shown the limitation of heatmap regression and proposed solution to tackle those limitation. For example, integral pose regression shows that it achieves similar numbers on MPII dataset, which the authors have not included in the table. However, the paper does show companion with pose regression for COCO dataset. Similarly [7] also shows similar performance on MPII dataset."*
>
> **A1:**
> Compared to previous works tackling the limitations of heatmap regression, our method outperforms these methods on MPII validation set.
>
> **(1) Comparison with integral pose regression.**
>
> MPII dataset has two subsets that can be used to assess performance, i.e., MPII validation set and MPII testing set. Integral pose regression published in ECCV 2018 reported its performance (mean PCKh\@0.5 score) on both MPII validation set (from 86.0 to 87.3) and MPII testing set (from 90.0 to 91.0). Since the ground-truth of MPII testing set is not publicly available, the evaluation on MPII testing set can only be done by sending an email to the dataset owner. Thus, most recent works ([39] from CVPR 2020, [15] from ICCV 2021, [7] from ICCV 2021, [a] from CVPR 2021, [b] from CVPR 2021) only evaluate their model performance on MPII validation set which we follow in our experiments. **On MPII validation set, the performance of our method (from 90.3 to 90.9) is significantly higher than the integral pose regression (from 86.0 to 87.3), showing the superior performance of our method.**
>
> Moreover, during our experiments, we also try to evaluate our method on MPII testing set by sending emails to the dataset owner. However, we did not get response. We will add integral pose regression to Tab. 3 of our paper and write the evaluation settings clearer in the revised version.
>
> **(2) Comparison with [7].**
>
> On MPII validation set, our performance is 0.4 higher than [7] on ResNet-152 and 0.3 higher than [7] on HRNet-W32. We believe that this level of performance improvement is already significant on MPII dataset. This is because in recent years, performance improvement on MPII dataset has been close to saturation, and other recent methods have also only yielded performance improvement at similar scales. For example, [7] (ICCV 2021) is 0.3 higher than its baseline on ResNet-152 and 0.2 higher than its baseline on HRNet-W32; [39] (CVPR 2020) is 0.3 higher than its baseline on HRNet-W32; [15] (ICCV 2021) is 0.1 higher than the previous method.
>
> Moreover, on the more challenging COCO dataset, our performance is significantly higher than [7] (76.7 vs 74.4 on COCO validation set with ResNet-152 as the backbone, 78.2 vs 75.8 on COCO validation set with HRNet-W32 as the backbone, and 77.2 vs 76.1 on COCO test-dev set with HRnet-W48 as the backbone).
>
> [a] Yu, Changqian, et al. Lite-HRNet: A lightweight high-resolution network. CVPR, 2021.
>
> [b] Li, Ke, et al. Pose recognition with cascade transformers." CVPR, 2021.

---

> > ### Author Response · Authors · 2022-08-02
> > **Response to reviewer wVS3's review (2/2)**
> >
> > >**Q2:** *"It will be interesting to observe how does this method compare to a heatmap regression based method where a multi-step training is applied where with each step of training the standard deviation is decreased. [...] If the objective is to decrease the area of localization it is important to observe a naive approach first. Several papers have shown that to work for facial landmark localization."*
> >
> > **A2:** Thanks for your suggestion. Below we compare our method with the multi-step training method where with each step of training the standard deviation is decreased. Specifically, a facial landmark localization work named "AdaLoss: Adaptive Loss Function for Landmark Localization" introduced two different methods of decreasing the standard deviation over steps. The first method (**Linear Decrease**) is to decrease the standard deviation linearly from a quarter of the heatmap resolution to 0 over epochs. The second method (**Decrease Based on Loss Variance**) is to decrease the standard deviation when the loss value has not changed significantly over the last 3 epochs. Below we compare our method with these methods and the baseline method on the same backbone model (HRNet-W48) on COCO validation set.
> >
> > | Method  | $AP$ | $AP^{50}$  | $AP^{75}$  | $AP^{M}$  | $AP^{L}$ | $AR$ |
> > |---|---|---|---|---|---|---|
> > | **Baseline**  | 77.1 | 91.8 | 83.8 | 73.5 | 83.5 | 81.8 |
> > | **Linear Decrease**  | 77.2 | 92.0 | 83.9 | 73.5 | 83.6 | 82.0 |
> > | **Decrease Based on Loss Variance**  | 77.3 | 92.1 | 84.0 | 73.6 | 83.6 | 82.0 |
> > | **Ours**  | **78.8** | **92.5** | **85.1** | **75.0** | **85.3** | **83.1** |
> >
> > As shown, the multi-step training methods (**Linear Decrease** and **Decrease Based on Loss Variance**) achieve slightly better performance than the baseline method. However, our performance is still significantly higher than the performance of these methods, demonstrating the effectiveness of our method that can consistently aggregate the pixel values in the predicted heatmap towards the dot annotation position and thus lead to superior performance.
> >
> > >**Q3:** *"It will be good to see the failure cases of this method. [...] Please provide failure cases in the experiment section. The paper does not talk about any limitations of the proposed approach."*
> >
> > **A3:** Thanks for the suggestion. We have shown the failure cases in Section E in the revised version of our Supplementary. As shown, in some extremely challenging cases (e.g., body joints under severe occlusion), both the baseline method and our method may not localize the body joints very accurately. This is also a limitation, and an important research problem in the task of pose estimation that we will take as a future direction.
> >
> > >**Q4:** *"Were the networks in the experiment initialized from already available pre-trained weights? If that in the case it will be interesting to observe how the networks train when trained from scratch. Do the networks converge with EMD when trained from scratch. [...] Earth movers distance can be difficult to train from scratch. This limits the paper to only be applied to already existing pre trained networks."*
> >
> > **A4:** Yes, the networks in the experiment are initialized from already available pre-trained weights. We also try to train networks from scratch with our method applied, and we report the results on COCO validation set below. Note that both methods below are with the same backbone model (HRNet-W48).
> >
> > | Method  | $AP$ | $AP^{50}$  | $AP^{75}$  | $AP^{M}$  | $AP^{L}$ | $AR$ |
> > |---|---|---|---|---|---|---|
> > | **Train from scratch**  | 78.8 | 92.4 | 85.3 | 75.0 | 85.2 | 83.2 |
> > | **Train from pre-trained weights**  | 78.8 | 92.5 | 85.1 | 75.0 | 85.3 | 83.1 |
> >
> > As shown, via training from scratch with our method applied, the networks can also converge and achieve similar performance. Thus, our method can be used both on models trained from scratch and models trained from pre-trained weights.

---

### Author Response · Authors · 2022-08-02
**General Response**

We thank all reviewers for recognition of our contributions (Reviewer wVS3:"an easy way to improve the quality of joint localization''; Reviewer 1LPW:"notable performance improvement''; Reviewer aExd:"novel and technically sound", "heatmap is a basic and important tool in human pose estimation'', "model-agnostic that can be applied in most pose estimator and improve the performance; Reviewer ggaN: "elegant and effective", "marvelous performance").

---

### Meta-Review · Area_Chair_Cpws · 2022-08-28

**Recommendation:** Accept
**Confidence:** Certain

**Metareview:**

This paper proposes to use earth mover distance to measure the loss function between a predicted heatmap and ground truth heatmap. It initially received mixed reviews. After rebuttal and discussion, all reviewers converged to acceptance of the paper. Reviewers believe this paper is novel and achieved significant practical performance across several models. AC follows the consensus and recommends acceptance of the paper.

**Award:**

No

---

### Decision · Program_Chairs · 2022-09-14

Accept